# Parallelly Running and Privacy-Preserving *k*-Nearest Neighbor Classification in Outsourced Cloud Computing Environments

Jeongsu Park  and Dong Hoon Lee *

Graduate School of Information Security, Korea University, Seoul 02841, Republic of Korea
* Correspondence: donghlee@korea.ac.kr

**Abstract:** Classification is used in various areas where *k*-nearest neighbor classification is the most popular as it produces efficient results. Cloud computing with powerful resources is one reliable option for handling large-scale data efficiently, but many companies are reluctant to outsource data due to privacy concerns. This paper aims to implement a privacy-preserving *k*-nearest neighbor classification (P*k*NC) in an outsourced environment. Existing work proposed a secure protocol (S*k*LE/S*k*SE) to compute *k* data with the largest/smallest value privately, but this work discloses information. Moreover, S*k*LE/S*k*SE requires a secure comparison protocol, and the existing protocols also contain information disclosure problems. In this paper, we propose a new secure comparison and S*k*LE/S*k*SE protocols to solve the abovementioned information disclosure problems and implement P*k*NC with these novel protocols. Our proposed protocols disclose no information and we prove the security formally. Then, through extensive experiments, we demonstrate that the P*k*NC applying the proposed protocols is also efficient. Especially, the P*k*NC is suitable for big data analysis to handle large amounts of data, since our S*k*LE/S*k*SE is executed for each dataset in parallel. Although the proposed protocols do require efficiency sacrifices to improve security, the running time of our P*k*NC is still significantly more efficient compared with previously proposed P*k*NCs.

**Keywords:** cloud computing; big data analysis; *k*-nearest neighbor classification; privacy-preserving computation



## 1. Introduction

In the era of big data, data mining and machine learning are important tools used to extract valuable information and predict outcomes, and these tools need to be able to analyze large-scale data [1,2]. For the sake of efficiency, large volumes of data are typically analyzed by cloud computing services at large IT companies, such as Amazon and Google [3], where there is ample and easy access to powerful resources for analyzing the plethora of data from a massive number of data owners. From the standpoint of data owners, it is more efficient to have the cloud handle analysis and return the results than attempt to analyze their own data. These days, as parallelly processing utilities, such as Hadoop, are becoming more widely disseminated, it is becoming easier to utilize cloud computing.

Although cloud computing for big data analysis has significant advantages, many companies and users are still reluctant to use these services due to privacy concerns surrounding outsourced cloud computing environments [3] because cloud computing service providers can access and reveal outsourced data, thus causing privacy problems. Even though data owners encrypt their own data before transmission as a preventative privacy-protecting measure, cloud service providers can obtain information regardless just by analyzing access patterns, which are the access records for original data according to computation results.

As for privacy protection techniques, there are secure multiparty computation and homomorphic encryption currently in use. In order to facilitate privacy-preserving computation for secret values in secure multiparty computation, non-colluding parties perform

computation with their shares generated from secret values in which the values or the computation results cannot be obtained by any party as long as the parties do not collude. However, since shared data are typically not encrypted, we focus on homomorphic encryption in this paper. Homomorphic encryption is an encryption scheme in which original data can be computed in an encrypted form by a third party, such as a cloud. Partially homomorphic encryption is a type of the homomorphic encryption and allows only one type of operation with an unlimited number of times [4].

Classification is a major data analysis task that is used in a variety of areas, such as medical diagnoses, spam mail detection, and credit evaluation [5–9]. In this vein, *k*-nearest neighbor classification is popular since it produces efficient results and yields high performance. Given a classified dataset and an unclassified input query, *k*-nearest neighbor classification selects *k* data most similar to the input query, which is classified by the majority class of the *k* data. The target of this present work is to implement a protocol to compute *k*-nearest neighbor classification privately, which we call privacy-preserving *k*-nearest neighbor classification (P*k*NC). Since *k*-nearest neighbor classification is used for big data analysis wherein the largest parameter is the number of data, it is critical that the communication round (i.e., running time) of P*k*NC is independent of the number of data.

As shown Figure 1, the proposed P*k*NC is executed in dual non-colluding cloud servers: a data host (DH) and a cryptographic service provider (CSP). DH receives an encrypted dataset from data owner and an encrypted input query from a querier, and CSP has a decryption key. At a high level, DH runs P*k*NC with CSP and then it returns the class of the input query based on the dataset in an encrypted form. Our P*k*NC does not disclose any information about a dataset, an input query, and the resultant class to an adversary as well as even DH and CSP to run P*k*NC.

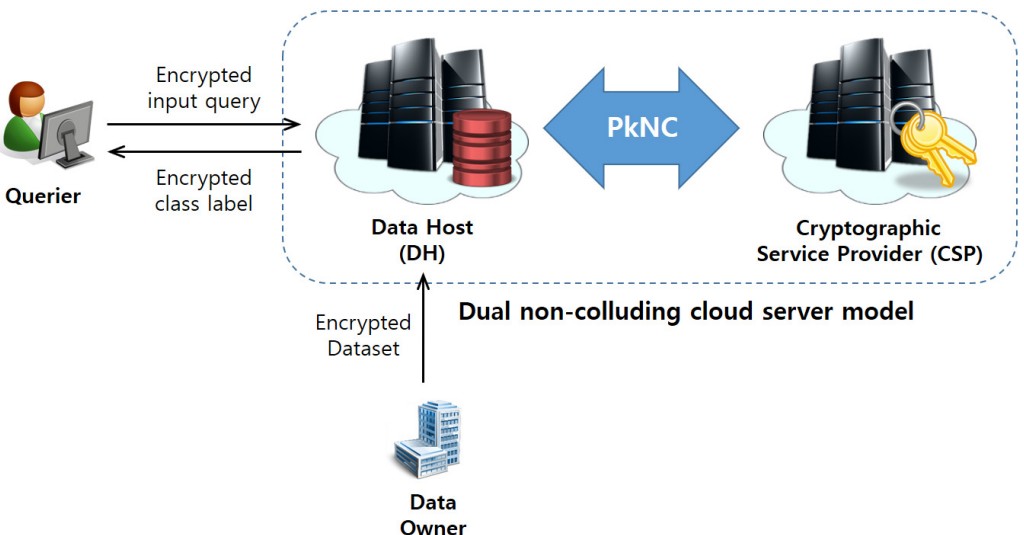

**Figure 1.** System model of P*k*NC.

In order to compute *k* data that are most similar to an input query (i.e., *k* smallest distances between the data and the input query) both privately and efficiently, one existing work [10] proposed a secure *k*-largest/smallest element (S*k*LE/S*k*SE) protocol and applied it to P*k*NC. The S*k*LE/S*k*SE executes at most *l* rounds where *l* is the length of an element, meaning that if *k* elements (distances) with the largest/smallest value are found before the last *l*-th round, it terminates for efficiency. The existence of different ending points for each input dataset implies that S*k*LE/S*k*SE [10] discloses information about the input dataset. In other words, starting from the $(l - 1)$-th bit of all input elements, the ending bit of S*k*LE/S*k*SE contains the information about *k* largest/smallest elements. For example, if S*k*LE to find *k* largest elements terminates in the first $(l - 1)$-th bit, the values of *k* largest elements are more than $2^{l-1}$. (This case is realistic since *l* is the effective size to represent

an element rather than the maximum size to support in a protocol.) Contrarily, if S$k$LE terminates in the last 0-th bit, two values of the smallest one in the $k$ largest elements and the largest one in the other elements are equal or their difference is 1.Therefore, it is necessary to protect the ending bit in existing S$k$LE/S$k$SE protocol [10]. Table 1 briefly shows the security of our proposed protocols compared with existing protocols.

**Table 1.** Security comparison of our protocols and existing protocols.

|           | Secure Comparison Protocol | S$k$LE/S$k$SE Protocol |
| :---: | :---: | :---: |
| [11]       | ✕ | ◯ |
| [10]       | ✕ | ✕ |
| [This work] | ◯ | ◯ |

Moreover, S$k$LE/S$k$SE requires a secure comparison protocol, but existing works [10,11] result in information disclosure problems. Specifically, when two input data are unequal, DH sends CSP a vector that consists of random values including 0 or 1. When two input data are equal, however, DH sends CSP a vector that consists of only random values, therefore meaning that CSP learns information about whether two input data are equal or not.

In addition, recently, researchers have proposed many P$k$NCs, but they are not formally proven [12–14] or expose some information about input data. P$k$NCs that are formally proven [11,15,16], unfortunately, are more inefficient as the volume of data increases, making these unsuitable for big data analysis that must compute large-scale data.

*Contributions*

In this paper, we propose new secure comparison and S$k$LE/S$k$SE protocols that solve the abovementioned information disclosure problems. We subsequently implement P$k$NC using the proposed protocols and demonstrate through the experiments that our proposals are practical. Firstly, we propose a secure comparison protocol that improves security by solving the information disclosure problem. In short, regardless of whether two input data are equal or unequal, DH sends CSP the similar vector that consists of either random values including 0 or only random values according to a random coin. Our secure comparison protocol guarantees privacy for the input data and results. We present this proposed secure comparison protocol in Section 4.1 and formally prove its security in Section 4.2.

Secondly, we propose a new S$k$LE/S$k$SE that improves security by solving the information disclosure problems in existing S$k$LE/S$k$SE [10]. To achieve this, the proposed S$k$LE/S$k$SE consistently terminates in the last round regardless of the input dataset, meaning that it does not disclose any information about the content of the input dataset. We denote the existing S$k$LE/S$k$SE [10] focused on efficiency by the efficient version of S$k$LE/S$k$SE (S$k$LE$_E$/S$k$SE$_E$). Similarly, we denote the proposed S$k$LE/S$k$SE to improve security by the secure version of S$k$LE/S$k$SE (S$k$LE$_S$/S$k$SE$_S$), which we present in Section 4.3. The proposed S$k$LE$_S$/S$k$SE$_S$ secures the privacy regarding the input dataset including the results and hides data access patterns even from DH and CSP. We formally prove its security in Section 4.4.

S$k$LE$_S$/S$k$SE$_S$ is advantageous because it is highly efficient for large dataset as it executes for each dataset in parallel. In other words, the communication round, which is proportionate to running time, is independent of the number of data, which indicates that it is suitable for big data analysis. It is additionally suitable for P$k$NC applications with a large $k$ of nearest neighbors since its communication round is independent of the parameter $k$. In order for existing protocols [11,17] to privately compute $k$ largest/smallest data in the dataset, they must serially run a maximum/minimum protocol to compare all data $k$ times. This means that the communication rounds in these existing protocols grow linearly with the number of data and parameter $k$. That is, existing works are unsuitable for both big data analysis and P$k$NC application with a large $k$.

In order to demonstrate that our proposed $SkLE_S$/$SkSE_S$ and secure comparison protocols are practical, we implement $PkNC$ including them and conduct extensive experiments with a real dataset. Figure 2 shows the ratio of the running time of $PkNCs$ for the same volume of data in which our $PkNC$ is much more efficient than existing $PkNCs$. Specifically, our $PkNC$ takes 4.38 min for 1728 data and 28.95 min for 8124 data. Note that the features of running time of our $PkNC$ is comparable to those of $SkLE_S$/$SkSE_S$, since the running time of $SkLE_S$/$SkSE_S$ accounts for most running time of our $PkNC$. The performance of our $PkNC$ is greatly improved in the cloud computing environment, since $SkLE_S$/$SkSE_S$ in our $PkNC$ is executed in parallel and the cloud enables numerous simultaneously running parallel operations. In addition, the running time of our $PkNC$ is also independent of $k$ of nearest neighbors like $SkLE_S$/$SkSE_S$. We present our $PkNC$ and its experiment in Section 5, where the experimental results support the above arguments.

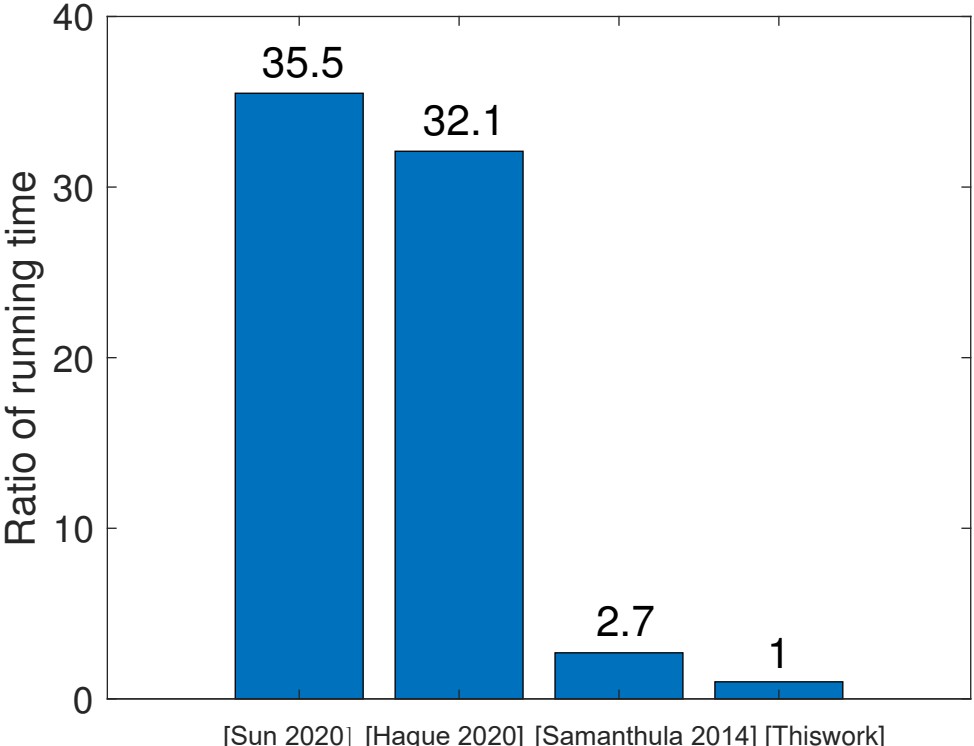

**Figure 2.** Running time ratio of $PkNCs$. Refs. [11,15,16] and [This work] respectively.

However, it cannot be denied that our security-enhanced protocols do sacrifice some efficiency. While the existing $SkLE_E$/$SkSE_E$ [10] runs at most $l$ rounds according to input dataset, our $SkLE_S$/$SkSE_S$ consistently runs $l$ rounds regardless, meaning that the number of communication rounds of $SkLE_S$/$SkSE_S$ is equal to or more than that of $SkLE_E$/$SkSE_E$. Our secure comparison protocol does require one more communication round than the existing comparison protocols [10,11]. Nevertheless, we emphasize that the improved security benefits of our proposed protocols compared to the existing protocols [10,11] outweigh this sacrifice, and our $PkNC$ is undeniably more efficient than existing $PkNCs$ [11,15,16]. Lastly, we summarize the contributions of this paper as follows.

- We propose a secure comparison (SCI) protocol to solve the information disclosure problem in existing works.
- Using the secure comparison, we propose new secure $k$-largest/smallest element ($SkLE$/$SkSE$) protocols, which solve the information disclosure problem and hide data access patterns.
- Using the proposed $SkLE$/$SkSE$, we implement a privacy-preserving $k$-nearest neighbour classification ($PkNC$) protocol.

- We prove the securities of the proposed protocols formally and demonstrate that the proposed protocols are practical through P$k$NC experiments with real datasets. In other words, P$k$NC is suitable for big data analysis to handle large-scale dataset and have large $k$ of nearest neighbors, since it is executed for each dataset in parallel.

The remainder of this paper is organized as follows. We briefly review existing works in Section 2 and explain preliminary concepts necessary for understanding our work such as system model and adversary model, performance evaluation measures, and functionalities for our proposed protocols in Section 3. In Section 4, we present the proposed secure comparison and S$k$LE$_S$/S$k$SE$_S$ protocols along with formal proofs. Then, we explain the implementation of P$k$NC using the proposed protocols and demonstrate their efficiency by analyzing experimental results in Section 5. Lastly, we conclude this work in Section 6.

## 2. Related Works

Privacy-preserving data analysis was first proposed by Lindell and Pinkas in 2000. In this protocol, there were two parties, each with their own confidential dataset, who wish to extract valuable information in union of their datasets without disclosing information to the other party. Since then, many researchers have become interested in privacy-preserving data analysis, especially P$k$NC, and have proposed many protocols related to P$k$NC, which became a hot issue.

The authors of [17] proposed a privacy-preserving $k$-nearest neighbor (PP$k$NN) using the Paillier cryptosystem with an additively homomorphic encryption property. The PP$k$NN guarantees privacy for both a dataset and an input query, including PP$k$NN results, and hides data access patterns. Once data owners outsource their datasets and a querier sends its query (as with P$k$NC), cloud servers (i.e., DH and CSP) do not need to communicate with the data owners or the querier. However, the PP$k$NN returns $k$ data closest to an input query rather than their majority class. The work of [11] improved on the PP$k$NN in [17] by proposing PP$k$NN classification (P$k$NC) to return the majority class of $k$ data closest to an input query as a result, which formally proved its security. However, the comparison protocol in the P$k$NC discloses information about whether two input data are equal, which we will explain in Section 4.1. We will also demonstrate that our P$k$NC is more efficient than the P$k$NC in [11] in Section 5.2.

The work of [18] proposed P$k$NC in an environment with multiple keys and multiple clouds. Similar to the existing works [11,17], this P$k$NC guarantees privacy of datasets and an input query along with a result and hides data access pattern. In this use of the P$k$NC, after data owners upload encrypted data to respective cloud server, they can download and decrypt the encrypted data since they encrypt the data with their own key. In order to run the P$k$NC, cloud servers first convert the data encrypted with their own key into the data encrypted with the same key by proxy re-encryption, but in doing so, the P$k$NC exposes class information of $k$ data closest to a query.

The authors of [19] proposed a more efficient P$k$NC than the scheme in [17] using Paillier and ElGamal cryptosystems. Similar to existing works, the P$k$NC returns the majority class of $k$ data closest to an input query as a result and provides privacy of a dataset, an input query, a result, and data access patterns. However, the P$k$NC exposes to a querier classes of the $k$ data closest to an input query rather than only their majority class. The authors in [20] proposed a very efficient P$k$NC for a large dataset, and this P$k$NC provides dataset security, key confidentiality, and query privacy as well as hides data access patterns. However, the P$k$NC does not provide semantic security for an outsourced dataset.

The authors of [21] proposed a very efficient PP$k$NN for a large dataset using an improved secure protocol for top-$k$ selection and proved its simulation-based security formally. However, in order to improve efficiency, the top-$k$ selection protocol returns an approximate result. In other words, it clusters a dataset using $k$-means algorithm and then, given a query, it selects several clusters that are closest to the query and computes the closest $k$ data in the clusters. Using the PP$k$NN to output an approximate result, though, is

unsuitable for applications that require an accurate classification result, such as medical diagnoses. The PP$k$NN also returns $k$ data closest to a query rather than the majority class.

The P$k$NC in [16] provides not only privacy but also reliability of collected data. By using Blockchain, it assures that a dataset collected by data owners is trustworthy. However, there are efficiency concerns with this protocol as this P$k$NC requires almost one hour to process only 760 data. In practice, our P$k$NC is much more efficient, which will be explained in detail later in this paper. There is another P$k$NC that was presented in [15], which is adaptable for a high-dimensional dataset. While most existing P$k$NCs deal only with integers, this P$k$NC allows a dataset and an input query to exist as real numbers. Similar to our P$k$NC, the running time of the P$k$NC is independent of $k$ of the nearest neighbors, but in contrast with our proposal, this P$k$NC requires huge memory to handle large volumes of data. The authors conducted an experiment for only 60 data in a machine with 8 GB RAM. This suggests that it requires large memory hardware, which is unsuitable for data analysis in the era of big data. Finally, the running time is also inefficient compared with our P$k$NC.

The authors of [22] proposed an efficient and privacy-preserving medical pre-diagnosis scheme based on multi-label $k$-nearest neighbors. Since a medical user can have multiple diseases at the same time, the scheme is practical. For the sake of efficiency, the scheme selects the dataset related to a medical user using $k$-means clustering and then performs the diagnosis scheme for the specific dataset. In other words, the scheme exposes data access patterns and cloud parties to run the scheme learn the information about a dataset or an input query. The work of [23] proposed PP$k$NN for eHealthcare data that combine kd-tree structure with homomorphic encryption. However, the PP$k$NN returns $k$ data closest to a query as a result rather than their majority class. Moreover, a user must be authorized by data owners before sending an input query and therefore, the scheme is impractical. The authors of [12] proposed P$k$NC using kd-tree technique and order-preserving encryption, which protects data privacy as well as data access patterns. However, since the scheme also assumes that data owners and users are honest, it is impractical and its application is limited.

PP$k$NN is used for location-based services. The scheme of [13] that utilizes Moore curve [24] protects the privacy of input data such as location information and ensures the accuracy of a query result. The authors of [25] proposed a verifiable PP$k$NN that uses network Voronoi diagram [26]. It not only ensures the confidentiality of input data but also verifies the integrity of results. The mechanism in [27] protects the location privacy of the Internet of Connected Vehicles using Intent-based Networking. Using the machine learning ability of the network, it predicts the intent of location accesses and penalizes the malicious access. The authors of [28] proposed a privacy-preserving data sharing scheme on the edge computing service of IoT, which provides data service for IoT devices. The privacy-preserving scheme based on attribute encryption scheme realizes anonymous data sharing and access control. The authors of [29] proposed an online privacy-preserving on-chain certificate status service based on the blockchain architecture, which ensures decentralized trust and provides privacy protection. In other words, the efficient privacy-preserving certificate status check protocol solves the problems of limited block size, high latency, and privacy leakage in comparison to existing works based on the blockchain technology.The work of [30] suggested a feature weighting algorithm to select an informative feature from redundant data. The feature weight is measured with the margin between the sample and its hyperplane, which is more robust to the noise and outliers than existing works.

## 3. Preliminaries

In this section, we introduce our system model, security definitions, and Paillier cryptosystem as an additively homomorphic encryption scheme. We also explain how to evaluate the performance of a protocol and briefly introduce the functionalities used in our protocols.

### 3.1. System Model

Our proposed protocols are executed in dual non-colluding cloud servers (Figure 1): data host (DH) and cryptographic service provider (CSP). CSP generates a public key for encryption and a secret key for decryption, then sends the public key to DH. DH, which already has encrypted input data, runs a protocol with CSP. After completing a protocol, DH returns a result in an encrypted form.

In dual non-colluding cloud server model, neither DH nor CSP disclose any information about input data or results. Specifically, since DH runs a protocol for data in an encrypted form, it actually cannot obtain any information about input data or results. Even though CSP decrypts encrypted intermediate results that it receives from DH, it cannot obtain any information about input data or the results since the decrypted data are blinded by a random value. Therefore, as long as DH does not collude with CSP, our protocols ensure that no information about input data or computation results are revealed. The dual non-colluding cloud server model is realistic and reasonable since large IT companies such as Amazon and Google provide cloud computing services that prioritize reputation over gains from collusion.

### 3.2. Adversary Model and Security Definitions

*Semi-Honest Adversary Model*: In this paper, we assume that DH and CSP operate within a semi-honest adversary model, in which a compromised party follows a protocol specification but tries to obtain information about an input data and results by analyzing intermediate results. For example, in comparison protocols of existing works [10,11], CSP obtains information about whether two input data are equal by decrypting and analyzing intermediate results received from DH. In $SkLE_E/SkSE_E$ [10], the information about an input dataset is also exposed by its end point. Creating a protocol in a semi-honest adversary model is a meaningful as the first step toward designing a protocol with stronger security.

*Security Definition*: In order to formally prove the security of our proposed protocols, we use the security definition of a semi-honest adversary model in terms of two-party computation [31]. Loosely speaking, we demonstrate that a simulator can generate the view of a corrupted party in real protocol execution when given only the input and output [32]. The view of a corrupted party consists of inputs, internal coin tosses, and received messages. If a simulator can generate indistinguishable values from the view of a corrupted party in real execution, then the definition states that the protocol is secure. The definition is as follows [31].

Let $f : \{0,1\}^* \times \{0,1\}^* \to \{0,1\}^* \times \{0,1\}^*$ be a functionality, and $f_{DH}(x,y)$ (resp., $f_{CSP}(x,y)$) denote DH's (resp., CSP's) element of $f(x,y)$. Let $\pi$ be a two-party protocol for computing $f$. DH's (resp., CSP's) view during an execution of $\pi$ on $(x,y)$, denoted $VIEW^{\pi}_{DH}(x,y)$ (resp., $VIEW^{\pi}_{CSP}(x,y)$), is $(x,r,m_1,\ldots,m_t)$ (resp., $(y,r,m_1,\ldots,m_t)$), where $r$ represents the outcome of DH's (resp., CSP's) internal coin tosses, and $m_i$ represents the $i$-th message it has received. DH's (resp., CSP's) output after an execution of $\pi$ on $(x,y)$, denoted $OUTPUT^{\pi}_{DH}(x,y)$ (resp., $OUTPUT^{\pi}_{CSP}(x,y)$), is implicit in the party's own view of the execution, and $OUTPUT^{\pi}(x,y) = (OUTPUT^{\pi}_{DH}(x,y),OUTPUT^{\pi}_{CSP}(x,y))$.

**Definition 1** (Privacy with respect to semi-honest behavior—general case)**.** *We say that $\pi$ privately computes $f$ if there exist probabilistic polynomial-time algorithms, denoted $S_{DH}$ and $S_{CSP}$ such that*

$$\{(S_{DH}(x,f_{DH}(x,y)),f(x,y))\}_{x,y} \equiv_c \{(VIEW^{\pi}_{DH}(x,y),OUTPUT^{\pi}(x,y))\}_{x,y} \qquad (1)$$

$$\{(S_{CSP}(y,f_{CSP}(x,y)),f(x,y))\}_{x,y} \equiv_c \{(VIEW^{\pi}_{CSP}(x,y),OUTPUT^{\pi}(x,y))\}_{x,y} \qquad (2)$$

$\equiv_c$ means that two distributions are computationally indistinguishable. Since the functionalities of our proposed protocols are probabilistic, we use the above general case security definition, with which we prove the security of our secure comparison protocol in Section 4.2 and that of $SkLE_S$ in Section 4.4.

*Sequential Modular Composition Theorem*: The sequential modular composition theorem [33] is a tool used to analyze the security of a protocol in a modular way [32]. We assume that $\pi_f$ is a protocol that computes a functionality $f$, which calls a subprotocol $\pi_g$ to compute a functionality $g$. The theorem states that, in order to analyze the security of $\pi_f$, it suffices to consider executing $\pi_f$ in a hybrid model where there is a third party to compute functionality $g$ ideally instead of a party that executes a real subprotocol $\pi_g$ [32]. Therefore, in order to analyze the security of a protocol in a modular way, one first proves the security of $\pi_g$ and then proves the security of $\pi_f$ in a model that allows a party to compute functionality $g$ ideally [32]. One denotes a model to analyze $\pi_f$ to call an ideal functionality $g$ instead of $\pi_g$ by $g$-hybrid model. We prove the security of our secure comparison protocol in the $F_{SZP}$-hybrid model in Section 4.2 and that of S$k$LE$_S$ in the $(F_{SM}, F_{SBD}, F_{SCI})$-hybrid model in Section 4.4.

### 3.3. Paillier Cryptosystem

As a partially homomorphic encryption scheme, we use the Paillier cryptosystem [34] in this paper. The Paillier cryptosystem is a probabilistic asymmetric encryption scheme with semantical security, which means that an adversary cannot learn any information about original data when given its encrypted data. Let $E_{pk}(\cdot) = E(\cdot)$ be the encryption function with a public key $pk$, and let $D_{sk}(\cdot) = D(\cdot)$ be the decryption function with a secret key $sk$, for which we drop the $pk$ and $sk$ for succinctness in this paper. The Paillier cryptosystem also holds additively homomorphic property which allows the addition of original data to be locally computed in an encryption form. In other words, given any two data $a, b \in \mathbb{Z}_N$, the following equations [10] are satisfied.

$$D(E(a) * E(b) \bmod N^2) = a + b \bmod N \tag{3}$$

$$D(E(a)^b \bmod N^2) = a \cdot b \bmod N \tag{4}$$

For succinctness, we drop the $\bmod N^2$ and the $\bmod N$ terms in the remainder of this paper. We stress that alternative additively homomorphic schemes can also be applied to our proposed protocols in lieu of the Paillier cryptosystem.

### 3.4. Performance Evaluation

We analyze the performance of a protocol in terms of computational costs (i.e., the number of encryptions/decryptions and exponentiations where we assume that encryption and decryption take the same amount of time) and communication costs (i.e., the amount of communication and the number of communication rounds) [10]. Since other operations other than encryption/decryption and exponentiation, such as homomorphic addition, have little influence on efficiency, we do not consider these in computational costs. The amount of communication means the total amount of transmitted data to complete a protocol, which we denote as a multiple of $C$ that is the size of a ciphertext. The number of communication rounds means the communication count executed in parallel [10].

### 3.5. Notation

For data $x$ with $0 \le x < 2^l$, we let $\langle x \rangle_B = \langle x_{l-1}, \ldots, x_1, x_0 \rangle$ by the binary representation of the data $x$, where $x_0$ (resp., $x_{l-1}$) is the least significant bit, denoted by LSB (resp., the most significant bit, denoted by MSB) and $x = \sum_{j=0}^{l-1} x_j \cdot 2^j$ for $x_j \in \{0, 1\}$ [10]. Similarly, for a ciphertext $E(x)$ with $0 \le x < 2^l$, we let $\langle E(x) \rangle_B = \langle E(x_{l-1}), \ldots, E(x_1), E(x_0) \rangle$ by ciphertexts for individual bits of corresponding data $x$, where $x = \sum_{j=0}^{l-1} x_j \cdot 2^j$ for $x_j \in \{0, 1\}$ [10]. Let $\overline{x}$ by 1's complement of data $x$, which is computed by toggling all bits of data. For example, 1's complement of binary number 1010 is 0101. Similarly, for a bit $x_i$, we let the complement of $x_i$ by $\overline{x_i}$, which is computed by $\overline{x_i} = 1 - x_i$ for $x_i \in \{0, 1\}$ [10].

$[n]$ for $n \ge 1$ means a set $\{1, 2, \ldots, n\}$. For a set $I = \{i_1, i_2, \ldots, i_n\}$, $\{d_i\}_{i \in I}$ means $\{d_{i_1}, d_{i_2}, \ldots, d_{i_n}\}$. $\{d_i\}_{i \in [l]}$ can be called a vector $d$. For a set $S$, $r \in_R S$ means that a value $r$ is chosen in the set $S$ uniformly at random. $DH \to CSP : E(x)$ means that $DH$ sends $CSP$ a

ciphertext $E(x)$. $a \cdot b$ means a multiplication operation in an integer and $E(a) * E(b)$ means a homomorphic addition mentioned in Section 3.3. Throughout this paper, we let the number of data by $n$, the size of the Paillier ciphertext by $C$, and the upper-bound number of bits required to represent data by $l$, which is less than or equal to the modulus size $|N|$ of the Paillier cryptosystem (i.e., $l \leq |N|$) [10].

### 3.6. Referenced Functionalities

In our protocols, calling a subprotocol to compute a functionality is presented as DH and CSP run an interactive protocol with a third party that computes the functionality ideally. Our proposed protocols call multiplication and bit decomposition protocols, for which we introduce secure multiplication functionality $F_{SM}$ and secure bit decomposition functionality $F_{SBD}$ in this subsection. The existing works [17,35] proposed the real protocols that privately compute $F_{SM}$ and $F_{SBD}$ in the dual non-colluding cloud server model mentioned in Section 3.1 and formally proved their security under semi-honest adversary model.

*Secure Multiplication functionality $F_{SM}$*: $F_{SM}$ receives $\{E(a), E(b)\}$ from DH and a secret key $SK$ from CSP, and then it sends $E(c)$ to DH where $c = a \cdot b$. We define $F_{SM}$ as follows.

$$F_{SM}(\{E(a), E(b)\}, SK) \rightarrow (E(c), \bot) \tag{5}$$

The real protocol that privately computes functionality $F_{SM}$ was proposed in [17]. It requires 6 encryptions/decryptions and 2 exponentiations, and $3 \cdot C$ bits are transmitted in 1 round.

*Secure Bit Decomposition functionality $F_{SBD}$*: $F_{SBD}$ receives $E(s)$ from DH and a secret key $SK$ from CSP, and then it sends $S'$ to DH where $S' = \{\langle E(s) \rangle_B, \langle E(\bar{s}) \rangle_B\}$. Recall that $\langle E(s) \rangle_B = \langle E(s_{l-1}), \ldots, E(s_1), E(s_0) \rangle$ and $\bar{s}$ is 1's complement of the data $s$. We define $F_{SBD}$ as follows.

$$F_{SBD}(E(s), SK) \rightarrow (S', \bot) \tag{6}$$

The real protocol that privately computes functionality $F_{SBD}$ is implemented by adding $E(\bar{s_i}) = E(1) * E(s_i)^{N-1}$ to the secure bit decomposition protocol proposed in [35]. We omit discussion about the detailed algorithm in this paper as it is trivial. This protocol requires $(3l + 1)$ encryptions/decryptions and $(4l + 2)$ exponentiations, and $(2l + 2) \cdot C$ bits are transmitted in $(l + 1)$ rounds.

## 4. Proposed Secure Comparison and S$k$LE/S$k$SE Protocols

As mentioned earlier, if the existing S$k$LE$_E$ [10] finds $k$ largest elements before the last $l$-th round, then it terminates. In other words, the existing S$k$LE$_E$ [10] exposes information about input dataset because the end points vary according to input data. In this section, we solve this information disclosure problem with our proposed S$k$LE$_S$/S$k$SE$_S$, whose end point is consistently the same regardless of input dataset, such that S$k$LE$_S$/S$k$SE$_S$ does not expose any information about the input dataset. In order to construct S$k$LE$_S$/S$k$SE$_S$, we first propose a secure comparison and inequality (SCI) protocol that does not disclose any information.

### 4.1. Secure Comparison and Inequality (SCI) Protocol

In this section, we propose an SCI protocol to compare two input data privately. The proposed SCI protocol solves the information disclosure problem that occurred in existing comparison protocols (SMIN of [11] and SCP of [10]), in which at a high level, when two input data are unequal (i.e., one input dataset is larger or smaller than the other), DH sends CSP a vector that consists of random values including 0 or 1. However, when two input data are equal, DH sends CSP a vector that consists of only random values; thus, CSP can learn information about whether the two input data are equal or not. Our proposed SCI protocol does not disclose any information about two input data since DH sends CSP a vector that consists of either random values including 0 or only random values according to a random coin when the two input data are equal as well as when the two input data are unequal.

The secure comparison and inequality functionality $F_{SCI}$ receives $\{S', k'\}$ from DH and a secret key $SK$ from CSP where $S' = \{\langle E(s)\rangle_B, \langle E(\bar{s})\rangle_B\}$ and $k' = \{\langle k\rangle_B, \langle \bar{k}\rangle_B\}$. Recall that $\langle E(s)\rangle_B = \langle E(s_{l-1}), \ldots, E(s_1), E(s_0)\rangle$ and $\bar{s}$ is 1's complement of the data $s$ as mentioned in Section 3.5. Then, $F_{SCI}$ sends $\{E(M), E(D)\}$ to DH where $E(M) = E(1)$ if $s < k$; otherwise, $E(M) = E(0)$ and $E(D) = E(1)$ if $s \neq k$ or else $E(D) = E(0)$. We define $F_{SCI}$ as follows.

$$F_{SCI}(\{S', k'\}, SK) \rightarrow (\{E(M), E(D)\}, \perp) \tag{7}$$

We present a real protocol to privately compute functionality $F_{SCI}$ in Algorithm 1 and provide an example in Table 2 for easy understanding. Our SCI returns not only a comparison result ($E(M)$) but also an inequality result ($E(D)$), and one of two input datasets is in plaintext form ($k'$). However, by modifying the SCI protocol slightly, it is possible to construct a common secure comparison protocol that returns only the comparison result without the inequality result, as well as ensuring that the two input data are all in an encrypted form. We omit discussion about the detailed algorithm because it is out of scope of this paper.

**Table 2.** Example of Algorithm 1 for SCI protocol ($l = 5$).

| Input | Functionality | j | $E(s_j)$ | $k_j$ | $E(w_j)$ | $E(x_j)$ | $E(y_j)$ | $E(\gamma)$ | $E(y_0)$ | $E(z_j)$ | $E(u_j)$ | $E(\beta)$ | $E(M)$ | $E(D)$ |
|---|---|---|---|---|---|---|---|---|---|---|---|---|---|---|
| (Case 1) $s < k$ | $F : s < k$ ($\alpha = 0$) | 4 | $E(1)$ | 1 | $E(0)$ | $E(0)$ | $E(0)$ | | | $E(-1)$ | $E(r)$ | | | |
| | | 3 | $E(1)$ | 1 | $E(0)$ | $E(0)$ | $E(0)$ | | | $E(-1)$ | $E(r)$ | | | |
| | | 2 | $E(0)$ | 1 | $E(r)$ | $E(1)$ | $E(1)$ | | | $E(0)$ | $E(r)$ | $E(1)$ | $E(1)$ | $E(1)$ |
| | | 1 | $E(1)$ | 0 | $E(0)$ | $E(1)$ | $E(r)$ | | | $E(r)$ | $E(r)$ | | | |
| | | 0 | $E(0)$ | 1 | $E(r)$ | $E(1)$ | $E(r)$ | $E(0)$ | $E(r)$ | $E(r)$ | $E(r)$ | | | |
| $s = 26$ $k = 29$ | $F : s \geq k$ ($\alpha = 1$) | 4 | $E(1)$ | 1 | $E(0)$ | $E(0)$ | $E(0)$ | | | $E(-1)$ | $E(r)$ | | | |
| | | 3 | $E(1)$ | 1 | $E(0)$ | $E(0)$ | $E(0)$ | | | $E(-1)$ | $E(r)$ | | | |
| | | 2 | $E(0)$ | 1 | $E(0)$ | $E(1)$ | $E(1)$ | | | $E(0)$ | $E(0)$ | $E(0)$ | $E(1)$ | $E(1)$ |
| | | 1 | $E(1)$ | 0 | $E(r)$ | $E(1)$ | $E(r)$ | | | $E(r)$ | $E(r)$ | | | |
| | | 0 | $E(0)$ | 1 | $E(0)$ | $E(1)$ | $E(r)$ | $E(0)$ | | $E(r)$ | $E(r)$ | | | |
| (Case 2) $s > k$ | $F : s < k$ ($\alpha = 0$) | 4 | $E(1)$ | 1 | $E(0)$ | $E(0)$ | $E(0)$ | | | $E(-1)$ | $E(r)$ | | | |
| | | 3 | $E(1)$ | 1 | $E(0)$ | $E(0)$ | $E(0)$ | | | $E(-1)$ | $E(r)$ | | | |
| | | 2 | $E(1)$ | 0 | $E(0)$ | $E(1)$ | $E(1)$ | | | $E(0)$ | $E(0)$ | $E(0)$ | $E(0)$ | $E(1)$ |
| | | 1 | $E(0)$ | 1 | $E(r)$ | $E(1)$ | $E(r)$ | | | $E(r)$ | $E(r)$ | | | |
| | | 0 | $E(1)$ | 0 | $E(0)$ | $E(1)$ | $E(r)$ | $E(0)$ | $E(r)$ | $E(r)$ | $E(r)$ | | | |
| $s = 29$ $k = 26$ | $F : s \geq k$ ($\alpha = 1$) | 4 | $E(1)$ | 1 | $E(0)$ | $E(0)$ | $E(0)$ | | | $E(-1)$ | $E(r)$ | | | |
| | | 3 | $E(1)$ | 1 | $E(0)$ | $E(0)$ | $E(0)$ | | | $E(-1)$ | $E(r)$ | | | |
| | | 2 | $E(1)$ | 0 | $E(r)$ | $E(1)$ | $E(1)$ | | | $E(0)$ | $E(r)$ | $E(1)$ | $E(0)$ | $E(1)$ |
| | | 1 | $E(0)$ | 1 | $E(0)$ | $E(1)$ | $E(r)$ | | | $E(r)$ | $E(r)$ | | | |
| | | 0 | $E(1)$ | 0 | $E(r)$ | $E(1)$ | $E(r)$ | $E(0)$ | | $E(r)$ | $E(r)$ | | | |
| (Case 3) $s = k$ | $F : s < k$ ($\alpha = 0$) | 4 | $E(1)$ | 1 | $E(0)$ | $E(0)$ | $E(0)$ | | | $E(-1)$ | $E(r)$ | | | |
| | | 3 | $E(1)$ | 1 | $E(0)$ | $E(0)$ | $E(0)$ | | | $E(-1)$ | $E(r)$ | | | |
| | | 2 | $E(0)$ | 0 | $E(0)$ | $E(0)$ | $E(0)$ | | | $E(-1)$ | $E(r)$ | $E(0)$ | $E(0)$ | $E(0)$ |
| | | 1 | $E(1)$ | 1 | $E(0)$ | $E(0)$ | $E(0)$ | | | $E(-1)$ | $E(r)$ | | | |
| | | 0 | $E(0)$ | 0 | $E(0)$ | $E(0)$ | $E(0)$ | $E(1)$ | $E(1)$ | $E(0)$ | $E(0)$ | | | |
| $s = 26$ $k = 26$ | $F : s \geq k$ ($\alpha = 1$) | 4 | $E(1)$ | 1 | $E(0)$ | $E(0)$ | $E(0)$ | | | $E(-1)$ | $E(r)$ | | | |
| | | 3 | $E(1)$ | 1 | $E(0)$ | $E(0)$ | $E(0)$ | | | $E(-1)$ | $E(r)$ | | | |
| | | 2 | $E(0)$ | 0 | $E(0)$ | $E(0)$ | $E(0)$ | | | $E(-1)$ | $E(r)$ | $E(1)$ | $E(0)$ | $E(0)$ |
| | | 1 | $E(1)$ | 1 | $E(0)$ | $E(0)$ | $E(0)$ | | | $E(-1)$ | $E(r)$ | | | |
| | | 0 | $E(0)$ | 0 | $E(0)$ | $E(0)$ | $E(0)$ | $E(1)$ | | $E(-1)$ | $E(r)$ | | | |

$r$: random value.

---

**Algorithm 1:** Secure Comparison and Inequality (SCI)

---

**DH input** : $\{S', k'\}$ where $S' = \{\langle E(s)\rangle_B, \langle E(\bar{s})\rangle_B\}$ and $k' = \{\langle k\rangle_B, \langle \bar{k}\rangle_B\}$

**CSP input** : secret key $SK$

**DH output**: $\{E(M), E(D)\}$ where $E(M) = \begin{cases} E(1), \text{ if } S < k \\ E(0), \text{ otherwise} \end{cases}$ and

$$E(D) = \begin{cases} E(1), \text{ if } S \neq k \\ E(0), \text{ otherwise} \end{cases}$$

**1 DH:**

**2** $\quad$ toss a random coin $\alpha \in_R \{0, 1\}$

**3** $\quad$ **for** $j \leftarrow l-1, \ldots, 0$ **do**

**4** $\quad\quad$ **if** $\alpha = 0$ **then**

**5** $\quad\quad\quad$ $E(w_j) \leftarrow E(\bar{s}_j)^{k_j}$

**6** $\quad\quad$ **else**

**7** $\quad\quad\quad$ $E(w_j) \leftarrow E(s_j)^{\bar{k}_j}$

**8** $\quad\quad$ **end**

**9** $\quad\quad$ $E(w_j) \leftarrow E(w_j)^{r_j}$ where $r_j \neq 0 \in_R \mathbb{Z}_N$

**10** $\quad\quad$ $E(x_j) \leftarrow E(s_j \oplus k_j)$

**11** $\quad\quad$ $E(y_j) \leftarrow E(y_{j+1})^{r'_j} * E(x_j)$ where $r'_j \neq 0 \in_R \mathbb{Z}_N$ and $E(y_l) = E(0)$

**12** $\quad$ **end**

**13 DH, CSP:**

**14** $\quad$ $(E(\gamma), \bot) \leftarrow F_{SZP}(\{E(x_j)\}_{j \in \{0, \ldots, l-1\}}, SK)$

**15 DH:**

**16** $\quad$ **if** $\alpha = 0$ **then**

**17** $\quad\quad$ $E(y_0) \leftarrow E(y_0) * E(\gamma)$

**18** $\quad$ **end**

**19** $\quad$ **for** $j \leftarrow l-1, \ldots, 0$ **do**

**20** $\quad\quad$ $E(z_j) \leftarrow E(y_j) * E(N-1)$

**21** $\quad\quad$ $E(u_j) \leftarrow E(z_j)^{r''_j} * E(w_j)$ where $r''_j \in_R \mathbb{Z}_N$

**22** $\quad$ **end**

**23** $\quad$ $\{v'_j\}_{j \in \{0, \ldots, l-1\}} \leftarrow \sigma(\{E(u_j)\}_{j \in \{0, \ldots, l-1\}})$

**24** $\quad$ $DH \rightarrow CSP : \{v'_j\}_{j \in \{0, \ldots, l-1\}}$

**25 CSP:**

**26** $\quad$ $\{v_j\}_{j \in \{0, \ldots, l-1\}} \leftarrow \{D(v'_j)\}_{j \in \{0, \ldots, l-1\}}$

**27** $\quad$ **if** $\exists v_j = 0 \text{ in } \{v_j\}_{j \in \{0, \ldots, l-1\}}$ **then**

**28** $\quad\quad$ $E(\beta) \leftarrow E(0)$

**29** $\quad$ **else**

**30** $\quad\quad$ $E(\beta) \leftarrow E(1)$

**31** $\quad$ **end**

**32** $\quad$ $CSP \rightarrow DH : E(\beta)$

**33 DH:**

**34** $\quad$ **if** $\alpha = 0$ **then**

**35** $\quad\quad$ $E(M) \leftarrow E(\beta)$

**36** $\quad$ **else**

**37** $\quad\quad$ $E(M) \leftarrow E(1) * E(\beta)^{N-1}$

**38** $\quad$ **end**

**39** $\quad$ $E(D) \leftarrow E(1) * E(\gamma)^{N-1}$

**40 return** $\{E(M), E(D)\}$

---

Intuitively, DH selects functionality $F(F : s < k$ or $F : s \geq k)$ by a random coin $\alpha$ and computes the functionality on two input data. CSP converts the computation result and returns the converted value ($\beta$) back to DH. Then, DH outputs result based on the converted value according to the random coin $\alpha$ (functionality $F$) selected by DH. As an idea for the SCI protocol, we modified the existing comparison protocol [10,11] so that an intermediate result of DH would be a vector that consists of either random values including 0 or only random values according to a random coin. We mentioned earlier that the intermediate result of DH in the existing comparison protocols [10,11] is either a vector that consists of random values including 0 or 1 if two input data are unequal or a vector that consists of only random values if two input data are equal. Therefore, when two input data are unequal, we modify 1 in a vector to be a random value. When two input data are equal, we modify one of the random values in a vector to 0 according to a random coin.

In order to avoid the scenario in which the intermediate result of DH becomes a vector that consists of only random values regardless of a random coin $\alpha$ when two input data are equal, we use functionality $F_{SZP}$, which privately computes whether all input data are 0 or not. Functionality $F_{SZP}$ receives $\{E(x_i)\}_{i \in [l]}$ from DH and a secret key SK from CSP, and then it sends $E(\gamma)$ to DH where $E(\gamma) = E(1)$ if all $x_i = 0$; otherwise, $E(\gamma) = E(0)$. We define $F_{SZP}$ as follows.

$$F_{SZP}(\{E(x_i)\}_{i \in [l]}, SK) \rightarrow (E(\gamma), \perp) \tag{8}$$

We present a real protocol to privately compute functionality $F_{SZP}$ in Algorithm A1 of Appendix A. The protocol requires $l$ encryptions/decryptions and $(l + 1)$ exponentiations, and $(l + 1) \cdot C$ bits are transmitted in 1 round.

For easier understanding of Algorithm 1, we intuitively explain data without encryption. DH selects functionality $F$ by tossing a random coin $\alpha$ (line 2), where $F : s < k$ if $\alpha = 0$; otherwise, $F : s \geq k$. When DH selects functionality $F : s < k$ (resp., $F : s \geq k$), $w_j$ is random if $(s_j, k_j) = (0, 1)$ (resp., $(s_j, k_j) = (1, 0)$); otherwise, $w_j = 0$ (lines 4–9). $x_j = 1$ if $s_j \neq k_j$; otherwise, $x_j = 0$ (line 10). $y_j = 1$ is in the first bit with $s_j \neq k_j$ from the $(l - 1)$-th bit and the other $y_j$ are either 0 or a random value (line 11). Let the first bit with $s_j \neq k_j$ from the $(l - 1)$-th bit be the $t$-th bit. Then $y_t = 1$, $y_j = 0$ for $j = t + 1, \ldots l - 1$, and $y_j$ is the random value for $j = 0 \ldots t - 1$.

If $s$ is equal to $k$, then $\gamma = 1$; otherwise $\gamma = 0$ (line 14), since $x_j = 0$ if $s_j = k_j$; otherwise, $x_j = 1$ in line 10. If DH selects $F : s < k$ ($\alpha = 0$), it adds $\gamma$ to $y_0$ so that it can send CSP a vector that consists of random values including 0 when $s = k$ and CSP returns $\beta = 0$ back to DH (lines 14–18). Note that, even though $\gamma$ is added to the fixed 0-th position of $y$, CSP cannot know the position since the information about the position is removed by a permutation $\sigma$ (line 23). If DH selects $F : S \geq k$ ($\alpha = 1$), it does not add $\gamma$ so that it can send CSP a vector that consists of only random values and CSP returns $\beta = 1$ back to DH.

Let the position with $y_j = 1$ be the $t$-th bit. When the selected functionality $F$ is different from the relation of the two input data, i.e., either when DH selects $F : s < k$, the relation of two input datasets is $s > k$ or when DH selects $F : s \geq k$, the relation of two input datasets is $s < k$—$u_t = 0$ and the other $u_j$ are random (lines 19–22). Conversely, when the selected functionality $F$ is same as the relation of two input data, i.e., either when DH selects $F : s < k$, the relation of two input datasets is $s < k$ or when DH selects $F : s \geq k$, the relation of two input datasets is $s > k$—all $u_j$ are random by adding $w_j$ in line 21. In addition, when two input data are equal ($s = k$) and the selected functionality is $F : s < k$, i.e., if the selected functionality $F$ is different from the relation of two input data—$u$ is a vector that consists of random values including $u_0 = 0$ since DH adds $\gamma = 1$ to $y_0$ in lines 16–18. If the selected functionality is $F : s \geq k$, i.e., when the selected functionality $F$ is same as the relation of two input data—$u$ is a vector that consists of only random values. After permutation $\sigma$ of $\{u_j\}_{j \in \{0, \ldots, l-1\}}$, DH sends CSP the permutated vector $v$ (lines 23–24).

CSP returns $\beta = 0$ (resp., $\beta = 1$) back to DH if it receives a vector that consists of random values including 0 (resp., a vector that consists of only random values). Specifically, CSP decrypts $v'_j$ and obtains $v_j$ (line 26). If there is an element with 0 in the vector $v$, CSP sends $\beta = 0$ to DH. If the vector $v$ consists of only random values, CSP sends $\beta = 1$ to DH

(lines 27-32). Even though the position of an element with 0 in a vector $u$ (line 21) includes information about the first position with $s_j \neq k_j$ or the 0-th bit when $s = k$, CSP cannot learn any information since it is removed by a permutation in line 23. DH privately computes the result $M$ based on the $\beta$ and according to the $\alpha$ selected by DH. Specifically, DH sets the result $M$ to $\beta$ ($M \leftarrow \beta$) if $\alpha = 0$ and the complement of $\beta$ ($M \leftarrow 1 - \beta$) if $\alpha = 1$ (lines 34–38). In conclusion, the result $M$ satisfies the condition $M = (s < k) = (\alpha \oplus \beta)$ for input data $s$ and $k$. DH privately computes inequality result $D$ for input data $s$ and $k$ (line 39).

*Computation and communication costs*: SCI protocol requires $2l$ encryptions/ decryptions and $(4l + 3)$ exponential computations, and $2(l + 1) \cdot C$ bits are transmitted in two rounds. Specifically, SZP requires $l$ encryptions/decryptions in line 14, and CSP decrypts $v'_j$ for $j = 0, \ldots, l - 1$ in line 26. Exponential computations are executed $3l$ times in lines 9, 11, and 21, and two times in lines 37 and 39. Exponential computations in lines 5 and 7 are excluded since $k_j$ is a public value in $\{0, 1\}$. Therefore, the total exponential computations of SCI are $(4l + 3)$ including $(l + 1)$ times in line 14. The amount of communication is $2(l + 1) \cdot C$ bits including $(l + 1) \cdot C$ bits in line 14. There are two communication rounds including once in line 14.

### 4.2. Proof of SCI Protocol

In this section, we denote SCI protocol (Algorithm 1) by $\pi_{SCI}$ and prove that $\pi_{SCI}$ privately computes $F_{SCI}$ in the $F_{SZP}$-hybrid model. In other words, we demonstrate that $\pi_{SCI}$ privately computes $F_{SCI}$ given access to functionality $F_{SZP}$. Intuitively, $\pi_{SCI}$ does not disclose any information about the comparison result including the input data to DH and CSP. In other words, the output $M$ of $\pi_{SCI}$ satisfies the condition $M = (s < k) = (\alpha \oplus \beta)$ where DH knows a random coin $\alpha$ and not $\beta$ since it receives the $\beta$ from CSP in an encrypted form. Therefore, DH cannot learn any information about the computation result. On the contrary, CSP knows $\beta$ and not $\alpha$ since CSP cannot learn any information about functionality $F$ selected by $\alpha$. This functionality is chosen uniformly at random by DH, and thus, CSP cannot learn any information about the computation result. In terms of proof, the DH simulator can generate the view of DH since DH only sees a random coin and data encrypted with semantically secure encryption scheme. The CSP simulator can also generate the view of CSP, which sees a vector that consists of either random values including 0 or only random values according to a random coin $\alpha$ which is selected uniformly at random.

**Theorem 1.** *$\pi_{SCI}$ privately computes $F_{SCI}$ in the $F_{SZP}$-hybrid model in the presence of a semi-honest adversary.*

**Proof of Theorem 1.** We demonstrate that joint distribution of the view and the output of the real protocol $\pi_{SCI}$ is computationally indistinguishable from that of the outputs of simulators and functionality $F_{SCI}$. Specifically, we demonstrate that (1) the view of DH is computationally indistinguishable from the output of the DH simulator, (2) the view of CSP is computationally indistinguishable from the output of the CSP simulator, and (3) the output of the real execution $\pi_{SCI}$ is computationally indistinguishable from that of functionality $F_{SCI}$.

(1) The view of DH in the real protocol $\pi_{SCI}$ is as follows.

$$VIEW_{DH}^{SCI}(\{S', k'\}, SK) = \{\{S', k'\}, \alpha, E(\gamma), E(\beta)\} \tag{9}$$

In $\pi_{SCI}$, $\alpha$ is a random coin that DH tosses (line 2), $E(\gamma)$ is a ciphertext returned from $F_{SZP}$ (line 14), and $E(\beta)$ is a ciphertext received from CSP (line 32). Intuitively, since $E(\gamma)$ and $E(\beta)$ are in an encrypted form, the DH simulator can generate the view as random values.

*DH simulator $S_{DH}$*
Input: The simulator $S_{DH}$ receives input $\{S', k'\}$ and output $\{E(M), E(D)\}$ of DH.
• Simulation

- The simulator chooses values $r^1$, $r^2$ and $r^3$ uniformly at random, where $r^1 \in_R \{0,1\}$ and $r^2, r^3 \in_R \mathbb{Z}_{N^2}$.
- The simulator defines $\{\{S', k'\}, r^1, r^2, r^3\}$ as the view of DH.
- The simulator outputs the view of DH and halts.

Random coin $\alpha \in_R \{0,1\}$ is indistinguishable from random $r^1 \in_R \{0,1\}$. Since the Paillier cryptosystem is semantically secure and the ciphertext is less than $N^2$, $E(\gamma)$ and $E(\beta)$ are computationally indistinguishable from $r^2$ and $r^3$. Therefore, the view of DH and the output of $S_{DH}$ are computationally indistinguishable.

(2) The view of CSP in the real protocol $\pi_{SCI}$ is as follows.

$$VIEW_{CSP}^{SCI}(\{S', k'\}, SK) = \{SK, \{v'_j\}_{j \in \{0,\dots,l-1\}}, \{v_j\}_{j \in \{0,\dots,l-1\}}\} \tag{10}$$

In $\pi_{SCI}$, $\{v'_j\}_{j \in \{0,\dots,l-1\}}$ is a vector that consists of ciphertexts received from DH (line 24) and $\{v_j\}_{j \in \{0,\dots,l-1\}}$ is obtained by decrypting the $\{v'_j\}_{j \in \{0,\dots,l-1\}}$ (line 26). Intuitively, the CSP simulator can generate $\{v_j\}_{j \in \{0,\dots,l-1\}}$, which is a vector that consists of either random values including 0 or only random values according to a random coin, and it can generate $\{v'_j\}_{j \in \{0,\dots,l-1\}}$ by encrypting the $\{v_j\}_{j \in \{0,\dots,l-1\}}$.

*CSP simulator $S_{CSP}$*
Input: The simulator $S_{CSP}$ receives a secret key $SK$ as CSP input.
• Simulation

- The simulator tosses a random coin $c \in \{0,1\}$.
- If a random coin $c$ is 0, the simulator sets $r_0^4$ to 0 and chooses $\{r_j^4\}_{j \in \{1,\dots,l-1\}}$ uniformly at random where $r_j^4 \in_R \mathbb{Z}_N$.
- If a random coin $c$ is 1, the simulator chooses $\{r_j^4\}_{j \in \{0,\dots,l-1\}}$ uniformly at random where $r_j^4 \in_R \mathbb{Z}_N$.
- The simulator permutes $\{r_j^4\}_{j \in \{0,\dots,l-1\}}$ uniformly at random and sets them to $\{r_j^5\}_{j \in \{0,\dots,l-1\}}$.
- The simulator computes $\{E(r_j^5)\}_{j \in \{0,\dots,l-1\}}$.
- The simulator defines $\{SK, \{E(r_j^5)\}_{j \in \{0,\dots,l-1\}}, \{r_j^5\}_{j \in \{0,\dots,l-1\}}\}$ as the view of CSP.
- The simulator outputs the view of CSP and halts.

As presented in Algorithm 1 ($\pi_{SCI}$), $\{v_j\}_{j \in \{0,\dots,l-1\}}$ is a vector that consists of either random values including 0 or only random values according to a random coin $\alpha$. Therefore, $\{v_j\}_{j \in \{0,\dots,l-1\}}$ is indistinguishable from $\{r_j^5\}_{j \in \{0,\dots,l-1\}}$. $\{v'_j\}_{j \in \{0,\dots,l-1\}}$ to encrypt $\{v_j\}_{j \in \{0,\dots,l-1\}}$ is indistinguishable from $\{E(r_j^5)\}_{j \in \{0,\dots,l-1\}}$.

(3) As explained earlier, the result of $\pi_{SCI}$ satisfies the condition $M = (\alpha \oplus \beta) = (s < k)$. As shown in Table 2, when $s < k$ ($M = 1$) and DH chooses a random coin $\alpha = 0$ (resp., $\alpha = 1$), then CSP returns $\beta = 1$ (resp., $\beta = 0$). On the contrary, when $s \geq k$ ($M = 0$) and DH chooses a random coin $\alpha = 0$ (resp., $\alpha = 1$), CSP returns $\beta = 0$ (resp., $\beta = 1$). Therefore, the result $E(M)$ of $\pi_{SCI}$ is the same as that of $F_{SCI}$. As for $E(D)$, $x_j = 0$ if $s_j = k_j$; otherwise, $x_j = 1$ (line 10). If $s$ is equal to $k$ (i.e., all $x_j$ is 0), $F_{SZP}$ returns $\gamma = 1$ (line 14) and $D = 0$ (line 39); otherwise if $s$ is unequal to $k$ (i.e., there is an element with 1 in a vector $x$), $F_{SZP}$ returns $\gamma = 0$ (line 14) and $D = 1$ (line 39). In other words, the result $E(D)$ of $\pi_{SCI}$ is the same

as that of $F_{SCI}$. Therefore, the output of $\pi_{SCI}$ is computationally indistinguishable from that of $F_{SCI}$. $\quad\square$

### 4.3. Secure Version of SkLE/SkSE (SkLE$_S$/SkSE$_S$)

*Secure version of SkLE (SkLE$_S$)*: In this subsection, we propose SkLE$_S$ to privately compute $k$ largest elements in an array (i.e., $k$ largest data in a dataset) in which no information is disclosed. The merit of SkLE$_S$ is that it is very efficient since it is executed for each element in parallel. In order to compute $k$ largest elements privately, the communication rounds of existing protocols [11,17] are proportional to the number of elements and parameter $k$ of nearest neighbors since they serially repeat maximum protocol $k$ times where the maximum protocol serially compares all elements. However, since our SkLE$_S$ is executed for each element in parallel and computes $k$ largest elements in only one execution, the communication rounds are independent of the number of elements and the parameter $k$. Therefore, it is suitable for both big data analysis that handles a large volume of data (elements) and P$k$NC applications with large $k$ of nearest neighbors. Since, for best performance, SkLE$_S$ needs to simultaneously execute as many operations as the number of elements, performance is greatly improved in the cloud computing environment, which enables numerous parallel operations. In addition, our SkLE$_S$ solves the information disclosure problem occurring in SkLE$_E$ [10]. The SkLE$_E$ [10] varies the end points running at most $l$ rounds according to input array where $l$ is the length of an element. This ultimately means that it discloses information about the input array. However, SkLE$_S$ consistently runs $l$ rounds regardless of input array, and therefore, it does not disclose any information about input array.

$F_{SkLE_S}$ receives a set of encrypted elements $\{\langle E(e_i)\rangle_B\}_{i\in[n]}$ from DH and a secret key $SK$ from CSP, and then it sends $\{E(K_i)\}_{i\in[n]}$ to DH where $K_i = 1$ if an element $e_i$ is included in the set of $k$ largest elements; otherwise, $K_i = 0$. We define $F_{SkLE_S}$ as follows.

$$F_{SkLE_S}(\{\langle E(e_i)\rangle_B\}_{i\in[n]}, SK) \rightarrow (\{E(K_i)\}_{i\in[n]}, \bot) \tag{11}$$

An element $e_i$ in an input dataset has auxiliary data that consists of $\{K_i, P_i, C_i\} \in \{0,1\}^3$, and SkLE$_S$ privately computes the set of $k$ largest elements by computing the auxiliary data for each round. $K_i$ is the output of SkLE$_S$, which indicates whether or not the corresponding element $e_i$ is included in the set of $k$ largest elements. $P_i$ means whether or not the corresponding element $e_i$ is a predicted $k$-largest element in corresponding round. SkLE$_S$ finds $k$ largest elements and a predicted $k$-largest element in the set of candidate elements, where $C_i$ indicates whether or not the corresponding element $e_i$ is included in the set of candidate elements. Once an element $e_i$ is included in the set of $k$ largest elements, it is irreversible. (i.e., $K_i = 0 \rightarrow 1$ but $1 \nrightarrow 0$). Similarly, once an element $e_i$ is excluded from the set of candidate elements, it is irreversible. (i.e., $C_i = 1 \rightarrow 0$ but $0 \nrightarrow 1$).

In each round, SkLE$_S$ privately computes auxiliary data for 1 bit of all elements from the $(l-1)$-th bit (MSB) to the 0-th bit (LSB) where $l$ is the length of an element. Resultant $k$ largest elements in an array are the elements $e_i$ with $K_i = 1$, which means the elements included in the set of $k$ largest elements. We present a real protocol to privately compute functionality $F_{SkLE_S}$ in Algorithm 2 and show an example in Table 3 for easy understanding. Note that DH locally performs all computations in Algorithm 2 except for the interactive protocols for functionalities $F_{SM}$, $F_{SBD}$, and $F_{SCI}$. Recall that $n$ is the number of all elements and $l$ is the upper-bound number of bits required to represent an element $e_i$.

The idea of SkLE$_S$ is to remove all elements from the set of candidate elements after it finds $k$ largest elements so that it keeps the results equal. Since the existing SkLE$_E$ [10] terminates after it finds $k$ largest elements, it does not need to consider computation of auxiliary data for $k$ largest elements. Since our SkLE$_S$ does not terminate after finding $k$ largest elements so that it is consistently executed $l$ rounds, it needs a method to keep the results of the $k$ largest elements equal even though it perform the same computation as before finding them. For this, when SkLE$_S$ finds $k$ largest elements, it removes all elements from the set of candidate elements ($C_i \leftarrow 0$) since $k$ largest elements are found in the set of candidate elements.

**Table 3.** Example of Algorithm 2 for S$k$LE$_S$ protocol.

| $j$ | $\{\langle E(e_i)\rangle_B\}_{i\in[5]}$ [1] | $\{E(P_i)\}_{i\in[5]}$ | $E(M)$ |
| | $\{E(K_i)\}_{i\in[5]}$ | $\{E(C_i)\}_{i\in[5]}$ | $E(D)$ |
|---|---|---|---|
| . | . | . | . |
| | $E(0), E(0), E(0), E(0), E(0)$ | $E(1), E(1), E(1), E(1), E(1)$ | . |
| 7 | $E(0), E(0), E(0), E(0), E(0)$ | $E(0), E(0), E(0), E(0), E(0)$ | $E(1)$ |
| | $E(0), E(0), E(0), E(0), E(0)$ | $E(1), E(1), E(1), E(1), E(1)$ | $E(1)$ |
| 6 | $E(1), E(0), E(0), E(0), E(0)$ | $E(1), E(0), E(0), E(0), E(0)$ | $E(1)$ |
| | $E(1), E(0), E(0), E(0), E(0)$ | $E(0), E(1), E(1), E(1), E(1)$ | $E(1)$ |
| 5 | $E(0), E(1), E(1), E(1), E(1)$ | $E(1), E(1), E(1), E(1), E(1)$ | $E(0)$ |
| | $E(1), E(0), E(0), E(0), E(0)$ | $E(0), E(1), E(1), E(1), E(1)$ | $E(1)$ |
| 4 | $E(0), E(1), E(0), E(0), E(0)$ | $E(1), E(1), E(0), E(0), E(0)$ | $E(1)$ |
| | $E(1), E(1), E(0), E(0), E(0)$ | $E(0), E(0), E(1), E(1), E(1)$ | $E(1)$ |
| 3 | $E(1), E(0), E(1), E(1), E(0)$ | $E(1), E(1), E(1), E(1), E(0)$ | $E(0)$ |
| | $E(1), E(1), E(0), E(0), E(0)$ | $E(0), E(0), E(1), E(1), E(0)$ | $E(1)$ |
| 2 | $E(0), E(1), E(1), E(0), E(1)$ | $E(1), E(1), E(1), E(0), E(0)$ | $E(0)$ |
| | $E(1), E(1), E(1), E(0), E(0)$ | $E(0), E(0), E(0), E(0), E(0)$ | $E(0)$ |
| 1 | $E(0), E(1), E(0), E(0), E(1)$ | $E(1), E(1), E(1), E(0), E(0)$ | $E(0)$ |
| | $E(1), E(1), E(1), E(0), E(0)$ | $E(0), E(0), E(0), E(0), E(0)$ | $E(0)$ |
| 0 | $E(1), E(0), E(1), E(1), E(0)$ | $E(1), E(1), E(1), E(0), E(0)$ | $E(0)$ |
| | $E(1), E(1), E(1), E(0), E(0)$ [2] | $E(0), E(0), E(0), E(0), E(0)$ | $E(0)$ |

Parameters : $k = 3$, $l = 8$; [1] input: $\{\langle E(e_i)\rangle_B\}_{i\in[5]} = \{\langle E(73)\rangle_B, \langle E(54)\rangle_B, \langle E(45)\rangle_B, \langle E(41)\rangle_B, \langle E(38)\rangle_B\}$; [2] output $\{E(K_i)\}_{i\in[5]} = \{E(1), E(1), E(1), E(0), E(0)\}$ means $\{73, 54, 45\}$ are the three largest elements in array $\{73, 54, 45, 41, 38\}$.

For easy understanding of Algorithm 2, we intuitively explain the data without encryption. First, DH initializes auxiliary data $K_i$ and $C_i$ for all elements $e_i$ so that there are no elements in the set of $k$ largest elements (i.e., $K_i \leftarrow 0$), and all elements are included in the set of candidate elements (i.e., $C_i \leftarrow 1$) (lines 2–5). For 1 bit of an element $e_i$, DH and CSP privately compute auxiliary data $P_i$, $K_i$, and $C_i \in \{0,1\}^3$ in each round (lines 6–24), which consists of the following four steps. We assume DH and CSP compute auxiliary data for the $j$-th bit of all elements in the $(l-j)$-th round ($j = l-1, \ldots, 0$).

---

**Algorithm 2:** Secure version of S$k$LE (S$k$LE$_S$)

---

**DH input** : set of encrypted elements $\{\langle E(e_i)\rangle_B\}_{i\in[n]}$

**CSP input** : secret key $SK$

**DH output**: $\{E(K_i)\}_{i\in[n]}$ where $K_i = 1$ if an element $e_i$ is included in the set of $k$ largest elements; otherwise, $K_i = 0$.

---

**1** DH locally performs all computations besides the interactive protocols for functionalities $F_{SM}$, $F_{SBD}$, and $F_{SCI}$.

**2** **for** $i \leftarrow 1, \ldots, n$ **do**

**3**    $E(K_i) \leftarrow E(0)$

**4**    $E(C_i) \leftarrow E(1)$

**5** **end**

**6** **for** $j \leftarrow l-1, \ldots, 0$ **do**

**7**    **for** $i \leftarrow 1, \ldots, n$ **do**

**8**       $(E(u_i), \bot) \leftarrow F_{SM}(\{E(e_{i,j}), E(C_i)\}, SK)$

**9**       $E(P_i) \leftarrow E(K_i) * E(u_i)$

**10**    **end**

**11**    $E(s) \leftarrow \prod_{i=1}^{n} E(P_i)$

**12**    $(S', \bot) \leftarrow F_{SBD}(E(s), SK)$

**13**    $(\{E(M), E(D)\}, \bot) \leftarrow F_{SCI}(\{S', k'\}, SK)$

**14**    $(E(\alpha), \bot) \leftarrow F_{SM}(\{E(D), E(M)\}, SK)$

**15**    $E(\beta) \leftarrow E(1) * E(D)^{N-1} * E(\alpha)$

**16**    $E(\gamma) \leftarrow E(D) * E(\alpha)^{N-2}$

**17**    **for** $i \leftarrow 1, \ldots, n$ **do**

**18**       $(E(v_i), \bot) \leftarrow F_{SM}(\{E(u_i), E(\beta)\}, SK)$

**19**       $E(K_i) \leftarrow E(K_i) * E(v_i)$

**20**       $(E(w_i), \bot) \leftarrow F_{SM}(\{E(e_{i,j}, E(\gamma)\}, SK)$

**21**       $E(x_i) \leftarrow E(M) * E(w_i)$

**22**       $(E(C_i), \bot) \leftarrow F_{SM}(\{E(C_i), E(x_i)\}, SK)$

**23**    **end**

**24** **end**

**25** **for** $i \leftarrow 1, \ldots, n$ **do**

**26**    $E(K_i) \leftarrow E(K_i) * E(C_i)$

**27** **end**

**28** **return** $\{E(K_i)\}_{i\in[n]}$

---

*(Step 1: lines 7–10) privately computing predicted k-largest element ($P_i$)*: A predicted $k$-largest element for the $j$-th bit is an element $e_i$ where bit 1 exists at least once from the $(l-1)$-th bit to the $j$-th bit. In other words, a predicted $k$-largest element for the $j$-th bit is either a candidate element ($C_i = 1$) whose $j$-th bit is 1 ($e_{i,j} = 1$) or a $k$-largest element ($K_i = 1$) in the previous round, which means that $e_{i,j} = 1$ exists at least once for $j = l-1, \ldots, j+1$. Therefore, $E(P_i)$ is computed as follows.

$$E(P_i) \leftarrow E(e_{i,j} \cdot C_i + K_i) \tag{12}$$

*(Step 2: line 11) privately computing the number of all predicted k-largest elements (s)*: Since the value of the auxiliary data for a predicted $k$-largest element is either 0 or 1 (i.e., $P_i \in \{0, 1\}$), the number $s$ of all predicted $k$-largest elements is computed by adding up all of the values as follow.

$$E(s) \leftarrow E(\textstyle\sum_{i=1}^{n} P_i) = \prod_{i=1}^{n} E(P_i) \tag{13}$$

*(Step 3: lines 12–13) privately comparing the number s of all predicated k-largest elements to parameter k of nearest neighbors*: For the comparison of $s$ and $k$, DH and CSP run an interactive protocol with a party to ideally compute functionality $F_{SCI}$ mentioned in Section 4.1. In

order to compute $S'$ for input of $F_{SCI}$, DH and CSP also run an interactive protocol with a party to ideally compute secure bit decomposition functionality $F_{SBD}$, as mentioned in Section 3.6. We do not present how to compute $k' = \{\langle k \rangle_B, \langle \overline{k} \rangle_B\}$ in this paper because $k$ is a public parameter and $k'$ can be computed easily.

*(Step 4: lines 14–23) privately computing k largest elements ($K_i$) and candidate elements ($C_i$):* As mentioned earlier, a predicted $k$-largest element is a candidate element whose bit is 1 in the corresponding round. Similarly, let unpredicted $k$-largest element be a candidate element whose bit is 0 in the corresponding round. DH and CSP privately compute whether or not an element ($e_i$) is included in the set of $k$ largest elements ($K_i$) and the set of candidate elements ($C_i$) according to comparison results ($M$ and $D$) of the number $s$ of predicted $k$-largest elements and parameter $k$. The idea to compute $K_i$ and $C_i$ is to include the predicted $k$-largest elements to the set of $k$ largest elements if $s < k$ and to exclude the unpredicted $k$-largest elements from the set of candidate elements if $s > k$. If $s = k$, all predicted $k$-largest elements are included in the set of $k$ largest elements, and the other candidate elements (i.e., unpredicted $k$-largest elements) are excluded from the set of candidate elements in order to keep the set of $k$ largest elements as a result since $k$ largest elements are found in the set of candidate elements.

Table 4 shows values of $C_i$ and $K_i$ for an element $e_i$ according to each case. (case 1) When $s < k$ (i.e., the number of predicted $k$-largest elements is less than parameter $k$), a predicted $k$-largest element (i.e., an element $e_i$ with $e_{i,j} = 1$ and in the set of candidate elements) is included in the set of $k$ largest elements ($K_i \leftarrow 1$) and is excluded from the set of candidate elements ($C_i \leftarrow 0$). (case 2) When $s > k$, an unpredicted $k$-largest element (i.e., an element $e_i$ with $e_{i,j} = 0$ that is in the set of candidate elements) is excluded from the set of candidate elements ($C_i \leftarrow 0$). (case 3) When $s = k$, a predicted $k$-largest element is included in the set of $k$ largest elements ($K_i \leftarrow 1$) and is excluded from the set of candidate elements ($C_i \leftarrow 0$). (case 4) Then, the other elements in the set of candidate elements (i.e., unpredicted $k$-largest elements) are excluded from the set ($C_i \leftarrow 0$). (case 5) Since there is no element in the set of candidate elements, the values of $K_i$ and $C_i$ for all elements (i.e., the set of $k$ largest elements and the set of candidate elements) are kept equal for the same computation of $K_i$ and $C_i$. According to Table 4, $E(K_i)$ and $E(C_i)$ are computed as follows.

$$E(K_i) \leftarrow E(K_i + e_{i,j} \cdot C_i \cdot (1 - D + D \cdot M)) \tag{14}$$

$$E(C_i) \leftarrow E(C_i \cdot (M + e_{i,j} \cdot D \cdot (1 - 2M))) \tag{15}$$

**Table 4.** Values of $K_i$ and $C_i$ according to the cases in S$k$LE$_S$.

| | Candidate $(K_i = 0, C_i = 1)$ | | Non-Candidate $(C_i = 0)$ | |
|---|---|---|---|---|
| | **Predicted** $(e_{i,j} = 1)$ | **Unpredicted** $(e_{i,j} = 0)$ | $e_{i,j} = 1$ | $e_{i,j} = 0$ |
| $s > k$ | $K_i \rightarrow K_i$ $C_i \rightarrow C_i$ | (case 2) $K_i \rightarrow K_i$ $C_i \rightarrow 0$ | | |
| $s < k$ | (case 1) $K_i \rightarrow 1$ $C_i \rightarrow 0$ | $K_i \rightarrow K_i$ $C_i \rightarrow C_i$ | (case 5) $K_i \rightarrow K_i$ $C_i \rightarrow C_i$ | |
| $s = k$ | (case 3) $K_i \rightarrow 1$ $C_i \rightarrow 0$ | (case 4) $K_i \rightarrow K_i$ $C_i \rightarrow 0$ | | |

When S$k$LE$_S$ cannot determine $k$ largest elements due to the presence of multiple elements with the same value, it returns as a result the elements in the set of candidate elements as well as the set of $k$ largest elements (lines 25–27). For example, when S$k$LE$_S$

finds the three largest elements ($k = 3$) in array $\{1, 2, 3, 3, 4, 5\}$, it returns the four largest elements $\{3, 3, 4, 5\}$ as a result.

*Parallelism*: In each round, the proposed S$k$LE$_S$ performs computation either for each element in parallel or all elements in common. Therefore, the communication rounds regarding running time is independent of the number of elements. The operations in lines 7–10 and lines 17–23 are computed for each element independently and in parallel. The operations in lines 12–16 are computed once in common regardless of the number of elements. Although just as many homomorphic additions are serially computed as the number of elements in line 11, we do not consider them as computation and communication costs since homomorphic addition has little influence on efficiency, as mentioned in Section 3.4.

*Computation and communication costs*: S$k$LE$_S$ requires $(24n + 5l' + 7) \cdot l$ encryptions/ decryptions and $(8n + 8l' + 9) \cdot l$ exponential computations, and $(12n + 4l' + 7) \cdot l \cdot C$ bits are transmitted in $(l' + 8) \cdot l$ rounds where $l$ represents the size of an element $e_i$, and $l'$ represents the size of the number of elements. Recall that SM requires six encryptions/decryptions and two exponential computations, and $3 \cdot C$ bits are transmitted in one round, as mentioned in Section 3.6. Specifically, SM in line 8 requires $6nl$ encryptions/decryptions and $2nl$ exponential computations, and $3nl \cdot C$ bits are transmitted since SMs for $n$ elements are repeated $l$ times. However, the number of communication rounds is $l$ since SMs for $n$ elements are executed in parallel. Likewise, SM operations in lines 18–22 are executed $nl$ times and the operations in lines 11–16 are executed $l$ times, respectively.

*Secure version of SkSE (SkSE$_S$)*: S$k$SE$_S$ privately computes $k$ smallest elements without disclosing information about an input array or results. In conclusion, S$k$SE$_S$ is constructed by inputting 1's complement of an input array to S$k$LE$_S$ as follows.

$$SkSE(\{\langle E(e_i)\rangle_B\}_{i\in[n]}) = SkLE(\{\langle E(\overline{e_i})\rangle_B\}_{i\in[n]}) \tag{16}$$

In order to construct S$k$SE$_S$, we followed a similar process to S$k$LE$_S$ in this section, and then reached the above conclusion. For more details, please refer to [10].

### 4.4. Proof of SkLE Protocol

In this section, we denote S$k$LE$_S$ protocol (Algorithm 2) by $\pi_{SkLE_S}$ and prove that $\pi_{SkLE_S}$ privately computes $F_{SkLE_S}$ in the $(F_{SM}, F_{SBD}, F_{SCI})$-hybrid model. Since S$k$SE$_S$ is constructed by S$k$LE$_S$, we do not prove the security of S$k$SE$_S$. We demonstrate that $\pi_{SkLE_S}$ privately computes $F_{SkLE_S}$ given access to the functionalities $F_{SM}$, $F_{SBD}$, and $F_{SCI}$. Intuitively, $\pi_{SkLE_S}$ does not disclose any information about an input array and results to DH and CSP since DH receives randomized ciphertexts from functionalities and CSP does not receive any data.

**Theorem 2.** *$\pi_{SkLE_S}$ privately computes $F_{SkLE_S}$ in the $(F_{SM}, F_{SBD}, F_{SCI})$-hybrid model in the presence of a semi-honest adversary.*

**Proof of Theorem 2.** We demonstrate that joint distribution of the view and the output of the real protocol $\pi_{SkLE_S}$ is computationally indistinguishable from that of the outputs of simulators and functionality $F_{SkLE_S}$. Specifically, we demonstrate that (1) the view of DH is computationally indistinguishable from the output of the DH simulator and (2) the output of DH in $\pi_{SkLE_S}$ is computationally indistinguishable from output of DH in $F_{SkLE_S}$. We do not consider the view and output of CSP because it does not receive any messages.

(1) We define DH's view of the real protocol $\pi_{SkLE_S}$ as follows.

$$VIEW_{DH}^{SkLE_S}(\{\langle E(e_i)\rangle_B\}_{i\in[n]}, SK) = \{\{\langle E(e_i)\rangle_B\}_{i\in[n]}, \{E(u_i)\}_{i\in[n]}, \{\langle E(s)\rangle_B, \langle E(\overline{s})\rangle_B\},$$
$$\{E(M), E(D)\}, E(\alpha), \{E(v_i)\}_{i\in[n]}, \{E(w_i)\}_{i\in[n]}, \{E(C_i)\}_{i\in[n]}\} \tag{17}$$

All data in DH's view are received from functionalities except for input $\{\langle E(e_i)\rangle_B\}_{i\in[n]}$. Specifically, $\{\langle E(s)\rangle_B, \langle E(\bar{s})\rangle_B\}$ are received from $F_{SBD}$ in line 12, $\{E(M), E(D)\}$ are received from $F_{SCI}$ in line 13, and the other data are received from $F_{SM}$. Intuitively, since DH sees the only data encrypted with a semantically secure encryption scheme, we can simulate DH's view.

*DH simulator $S_{DH}$*
Input: The simulator $S_{DH}$ receives input $\{\langle E(e_i)\rangle_B\}_{i\in[n]}$ and output $\{E(K_i)\}_{i\in[n]}$ of DH.
• Simulation
- The simulator $S_{DH}$ chooses values $\{r_i^1\}_{i\in[n]}$, $\{r_i^2\}_{i\in\{0,\ldots,l-1\}}$, $\{r_i^3\}_{i\in\{0,\ldots,l-1\}}$, $r^4$, $r^5$, $r^6$, $\{r_i^7\}_{i\in[n]}$, $\{r_i^8\}_{i\in[n]}$, $\{r_i^9\}_{i\in[n]}$ uniformly at random where the values are in $\mathbb{Z}_{N^2}$.
- The simulator defines $\{\{\langle E(e_i)\rangle_B\}_{i\in[n]}, \{r_i^1\}_{i\in[n]}, \{\{r_i^2\}_{i\in\{0,\ldots,l-1\}}, \{r_i^3\}_{i\in\{0,\ldots,l-1\}}\}, \{r^4, r^5\}, r^6, \{r_i^7\}_{i\in[n]}, \{r_i^8\}_{i\in[n]}, \{r_i^9\}_{i\in[n]}\}$ as the view of DH.
- The simulator outputs DH's view and halts.

$\{E(u_i)\}_{i\in[n]}$ are computationally indistinguishable from $\{r_i^1\}_{i\in[n]}$ since the Paillier cryptosystem is semantically secure and its ciphertext is less than $N^2$. Similarly, the other data in the view of DH are computationally indistinguishable from the simulator's outputs. Therefore, the distribution of DH's view is computationally indistinguishable from outputs of the simulator $S_{DH}$.

(2) Let $\pi_{SkLE_S}$ compute auxiliary data for the $j$-th bit of all elements in the $(l-j)$-th round ($j = l-1, \ldots, 0$). It is clear that a predicted $k$-largest element will be larger than other elements for the $j$-th bit because, from $(l-1)$-th bit to $j$-th bit, a predicted $k$-largest element has bit 1 at least once but the other elements are all 0. Predicted $k$-largest elements include $k$ largest elements in the corresponding round. As shown in case 1 of Table 4, when $s < k$, predicted $k$-largest elements are included to the set of $k$ largest elements ($K_i \leftarrow 1$). The prior $k$-largest elements remain the same since the predicted $k$-largest elements include them. When $s = k$, predicted $k$-largest elements are included in the set of $k$ largest elements ($K_i \leftarrow 1$) as shown in Case 3. In addition, the other candidate elements (i.e., unpredicted $k$-largest elements) are excluded from the set of candidate elements ($C_i \leftarrow 0$) as shown in Case 4, so that $k$ largest elements remain the same as in case 5. Since all elements in the set of $k$ largest elements are larger than the other elements, output of $\pi_{SkLE_S}$ is computationally indistinguishable from that of functionality $F_{SkLE_S}$.

*Hiding data access patterns*: $SkLE_S$ hides data access patterns for $k$ largest ones of all elements. Informally, $SkLE_S$ performs either the same computation for each element or a common computation for all elements. Specifically, auxiliary data $P_i$, $K_i$, and $C_i$ regarding $k$ largest elements are computed by the same equations regardless of result (lines 7–10 and lines 17–23). The other computations in lines 11–16 are executed for all common elements. Since all data are encrypted with semantically secure encryption schemes, and therefore no information is disclosed, $SkLE_S$ is secure against data access pattern attacks. □

## 5. Implementation and Experimental Results of Privacy-Preserving $k$-Nearest Neighbor Classification

In order to demonstrate the efficiency, we implemented a privacy-preserving $k$-nearest neighbor classification (P$k$NC) using the proposed $SkLE_S$/$SkSE_S$ and SCI protocols. Our extensive experiments contained real datasets, and we compared the experimental results with the results from existing P$k$NC experiments [11].

### 5.1. Privacy-Preserving k-Nearest Neighbor Classification

Given an unclassfied input query and a classified dataset, $k$-nearest neighbor classification selects $k$ data most similar to the input query and classifies the unclassified input query by the majority class of the $k$ selected data. Typically, P$k$NC algorithm consists of the following three steps.

- Step 1: computing distances between an input query and data in a dataset.
- Step 2: selecting $k$ smallest distances.
- Step 3: computing the majority class of $k$ data corresponding to the $k$ smallest distances.

Table 5 shows the ratio of running time broken down by steps in our P$k$NC when applying S$k$LE$_S$/S$k$SE$_S$. Since the running time of S$k$LE$_S$/S$k$SE$_S$ accounts for most of the P$k$NC running time, the features of S$k$LE$_S$/S$k$SE$_S$ lead to those of P$k$NC in terms of running time. In order to improve the efficiency of P$k$NC, it is significant to compute step 2 efficiently. For more details about our P$k$NC algorithm, refer to Algorithm A2 in Appendix B.

**Table 5.** Running time ratio by steps in our P$k$NC

|  | Step 1 | Step 2 | | Step 3 | Total |
|---|---|---|---|---|---|
|  |  | SBD | S$k$LE$_S$ |  |  |
| Ratio | 5 % | 9 % | 75 % | 11 % | 100 % |

As mentioned earlier, S$k$LE/S$k$SE has an efficient version (S$k$LE$_E$/S$k$SE$_E$) that focuses on efficiency and a secure version (S$k$LE$_S$/S$k$SE$_S$) that improves security. When comparing the secure version and the efficient version in terms of communication rounds related to running time, the secure version requires $2l$ communication rounds more than the efficient version in the worst case scenario. Despite this, we emphasize again that the security of the secure version is much improved.

*5.2. Implementation and Experimental Results*

We implemented P$k$NC to apply S$k$LE$_S$/S$k$SE$_S$ and SCI using the Paillier cryptosystem [36] as an additively homomorphic encryption scheme in C++. Then, we conducted an experiment on two Linux machines for DH and CSP. The Linux system features an Intel Core i7-4790 CPU 3.60 GHz processor and 15 GB RAM running Ubuntu 18.04 LTS. In particular, the machine has four cores and runs eight parallel operations via hyper-threading technology [37].

In order to show performance of our P$k$NC applying the proposed S$k$LE$_S$/S$k$SE$_S$ clearly, we conducted experiments with the Car Evaluation dataset used in existing work [11] and the Mushroom dataset from the UCI machine learning repository [38]. The Car Evaluation dataset [39] consists of 1728 data with six attributes and four classes, and the Mushroom dataset [40] consists of 8124 data with twenty-two attributes and two classes.

We first ran our P$k$NC for $k = 1000$ with the Car Evaluation dataset, which took 4 min 23 s when the key size is 1024 bits and the number of threads is 8. Table 6 compares running times of our P$k$NC and existing P$k$NCs. These results verify that our P$k$NC, when utilizing S$k$LE$_S$/S$k$SE$_S$ and SCI protocols, is very efficient. Starting the above experiment, we varied the parameters such as the $k$ of nearest neighbors, key size, the number of data, and the number of threads for parallel operations. At last, we compared and analyzed the resultant running times to that of existing P$k$NC [11], which is the most efficient of the previous protocols in Table 6.

**Table 6.** Running time comparison of our P$k$NC and the existing P$k$NCs.

|  | Number of Data | Running Time |
|---|---|---|
| [16] | 760 | 61.8 min |
| [15] | 60 | 5.4 min |
| [11] | 1728 | 12 min |
| [This work] | 1728 | 4.38 min |

Figure 3 shows the running times of our P$k$NC applying S$k$LE$_S$/S$k$SE$_S$ and an existing P$k$NC [11] for the $k$ of nearest neighbors. It shows that the running time of our P$k$NC is independent of $k$ as S$k$LE$_S$/S$k$SE$_S$. Since the existing P$k$NC runs a minimum protocol $k$

times in order to privately compute $k$ data with the smallest value, its running time rapidly increases for parameter $k$, but since $SkLE_S/SkSE_S$ in our P$k$NC privately computes $k$ smallest data via only one execution regardless of $k$, our P$k$NC is independent of parameter $k$. Specifically, the existing P$k$NC took 12.02 min to 55.5 min as $k$ increases from 5 to 25, and we expect that it would take more than one hour for $k$ values exceeding 25. However, our P$k$NC took roughly 4.4 min regardless of parameter $k$. Therefore, our P$k$NC is much more efficient for $k$ than the existing P$k$NC thanks to the efficiency of $SkLE_S/SkSE_S$.

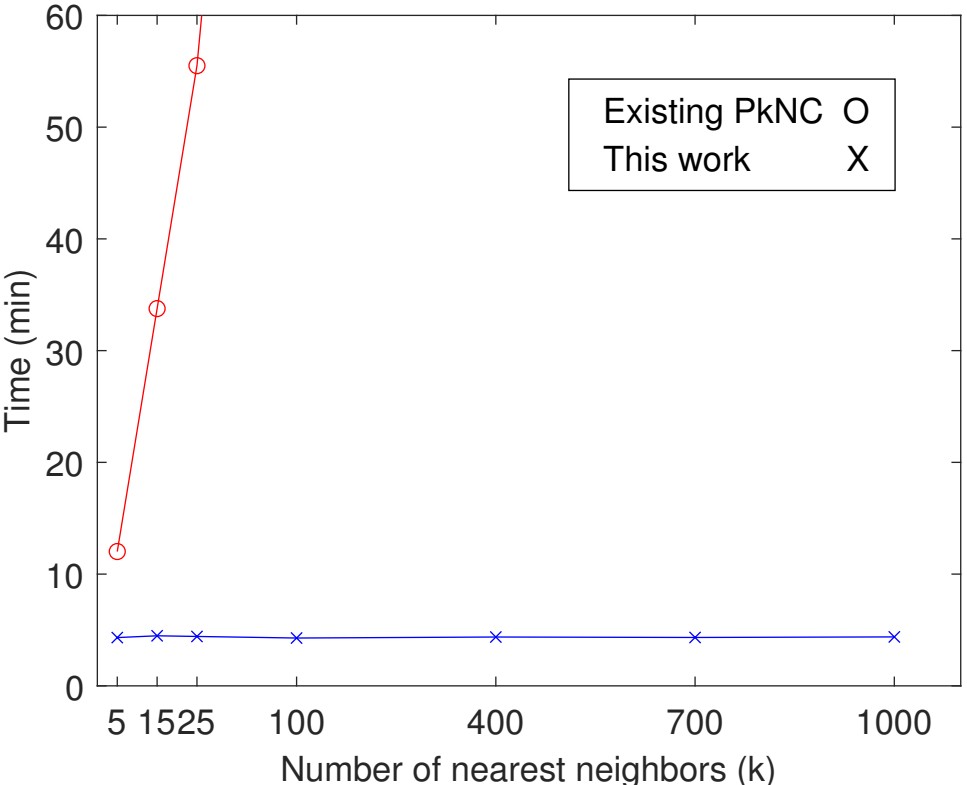

**Figure 3.** Running time comparison of our P$k$NC and the existing P$k$NC [11] for the number of nearest neighbors ($k$).

Figure 4 shows the running time of our P$k$NC applying $SkLE_S/SkSE_S$ and the existing P$k$NC [11] for key size. It shows that the running time of our P$k$NC gradually increases in comparison with the existing P$k$NC. As the key size increases from 512 bits to 1024 bits, while running time of the existing P$k$NC rapidly increases 9.98 min to 66.97 min, our P$k$NC increases 1.2 min to 4.38 min. For 2048 bits of key size, our P$k$NC took only 27.2 min. Therefore, our P$k$NC is more efficient than the existing P$k$NC for key size.

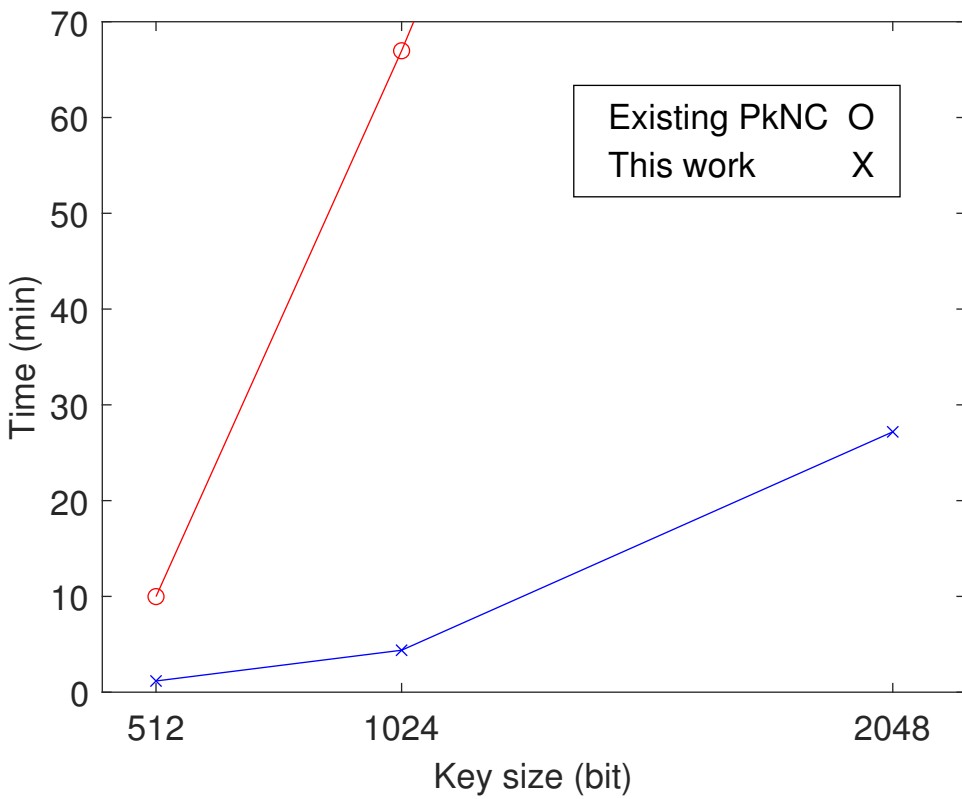

**Figure 4.** Running time comparison of our P$k$NC and the existing P$k$NC [11] for key size (bit).

Table 7 illustrates the amount of communication for our P$k$NC versus an existing P$k$NC [11] when key size is 1024 bits. It shows that the communication amount of our P$k$NC is roughly one-third of that of the existing P$k$NC. Specifically, while the existing P$k$NC needs to transmit data of 154.78 megabytes, our P$k$NC transmits only 54.72 megabytes. By assuming a common 10 Mbps LAN environment, while the network delay for the existing P$k$NC was 123.82 s, the delay for our P$k$NC is only 44 s.

**Table 7.** Communication amount and network delay of our P$k$NC and the existing P$k$NC [11] in 10 Mbps LAN.

|  | Communication Amount (Megabytes) | Network Delay | (s) |
|---|---|---|---|
| [11] | 154.78 | 123.82 | |
| [This work] | 54.72 | 43.77 | |

We also conducted an experiment for much more data than the Car Evaluation dataset ($n = 1728$) used in existing P$k$NC [11]. Even though the number of data in the Mushroom dataset ($n = 8124$) is more than the Car Evaluation dataset by 4.7 times, the running time of our P$k$NC is less than 30 min, which is more efficient than the existing P$k$NC with $k = 15$. Therefore, our P$k$NC is more efficient than the existing P$k$NC for the number of data.

Figure 5 shows the running time of our P$k$NC for the number of threads (parallel operations). It implies that the performance of our P$k$NC is highly improved in the cloud computing environment to enable numerous parallel operations. This is because the proposed S$k$LE$_S$/S$k$SE$_S$ is executed for each dataset in parallel, as mentioned in Section 4.3. In other words, the figure shows that the running time decreased by half as the number of threads doubled. Specifically, our P$k$NC took 23.5 min for only one thread without parallel operations. When the number of threads doubled, its running time decreased

roughly by half. That is, for our experiment, it took 11.4 min for two threads. Likewise, when the number of threads doubled, it took 5.3 min for four threads. However, when the number of threads reached eight, its running time decreased by slightly less than half since our machines used in the experiment have four cores and allow eight parallel operations via hyper threading technology. For more than eight threads, the running time ceased to decrease since our machines allow for at most eight parallel operations. This implies that performance of our P$k$NC, including S$k$LE$_S$/S$k$SE$_S$, can be improved greatly if it is executed in the cloud, where more parallel operations are allowed and supported.

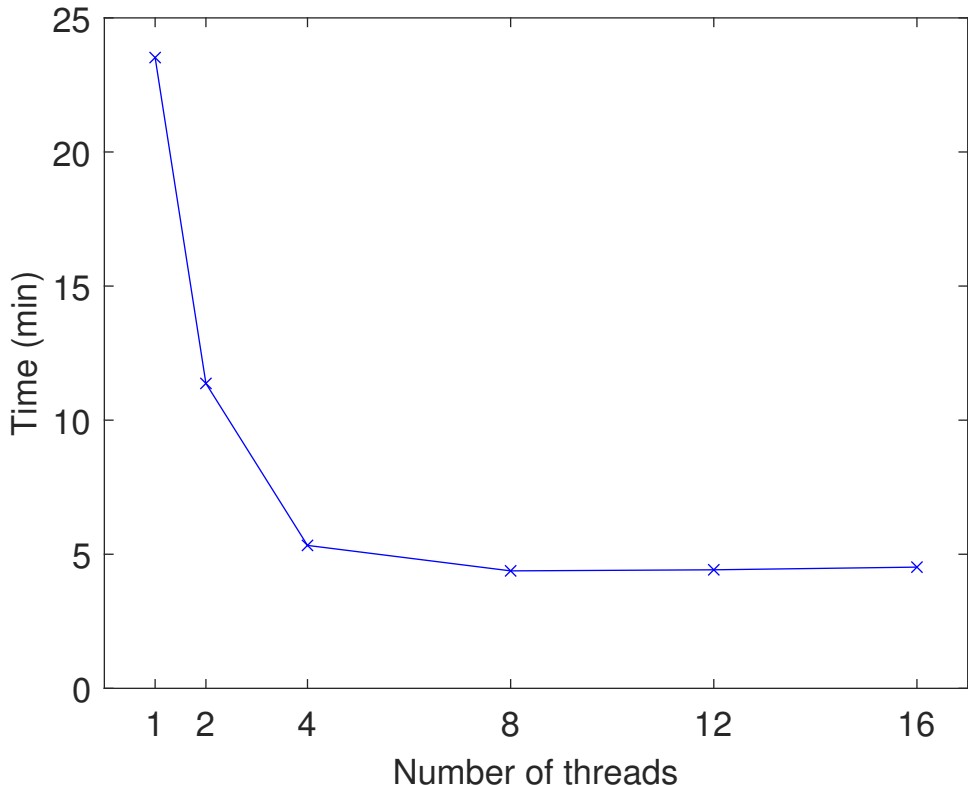

**Figure 5.** Running time for the number of threads.

However, our P$k$NC with S$k$LE$_S$/S$k$SE$_S$ applied requires a little more running time than the existing P$k$NC [10] with S$k$LE$_E$/S$k$SE$_E$, which focuses on efficiency but reveals some information. By our observations, the running time of our P$k$NC roughly increased by 11% in comparison with the existing P$k$NC [10]. The first reason for this is that S$k$LE$_S$/S$k$SE$_S$ requires 2l communication rounds more than S$k$LE$_E$/S$k$SE$_E$ as mentioned in Section 5.1. The second reason is that our S$k$LE$_S$/S$k$SE$_S$ consistently terminates in the last round regardless of an input dataset, while the earlier S$k$LE$_E$/S$k$SE$_E$ can terminate before the last round, or in the worst case scenario, it terminates in the last round according to an input dataset. In other words, the number of rounds for the proposed S$k$LE$_S$/S$k$SE$_S$ is more than or equal to that of the earlier S$k$LE$_E$/S$k$SE$_E$. The last reason is that our SCI protocol requires one more communication round than the existing comparison protocols [10,11]. However, we emphasize that the security of our S$k$LE$_S$/S$k$SE$_S$ is improved in comparison to that of S$k$LE$_E$/S$k$SE$_E$. Therefore, S$k$LE$_S$/S$k$SE$_S$ and S$k$LE$_E$/S$k$SE$_E$ should be selected in regards to the trade-off between security and efficiency. Nevertheless, we emphasize that the running time of our P$k$NC with S$k$LE$_S$/S$k$SE$_S$ applied is even more efficient than that of the existing P$k$NCs [11,15,16].

## 6. Conclusions

Data mining and machine learning are highly significant tools necessary for the analysis of large-scale data to return meaningful information. In order to handle large volumes of data efficiently, using outsourced cloud computing services has emerged as a viable option but that can lead to privacy problems, which is a pressing concern to resolve. Therefore, we focused on a privacy-preserving k-nearest neighbor classification (P$k$NC) in outsourced cloud computing environment. To this end, we proposed S$k$LE$_S$/S$k$SE$_S$ and SCI protocols to solve the information disclosure problems of S$k$LE$_E$/S$k$SE$_E$ and secure comparison protocols in the existing works [10,11]. We formally proved the securities of our S$k$LE$_S$/S$k$SE$_S$ and SCI protocols via the simulation paradigm. Then, we implemented P$k$NC to apply the proposed protocols in C++ and conducted extensive experiments with real datasets. Although the proposed S$k$LE$_S$/S$k$SE$_S$ and SCI protocols sacrifice some efficiency to improve security, our P$k$NC with the relevant protocols applied is still more efficient than the existing P$k$NCs.

The efficient and private algorithms, such as S$k$LE$_S$/S$k$SE$_S$ that are executed parallelly and disclose no information in outsourced cloud computing environments, will play an important role in improving the efficiency of big data analysis. We will continue, therefore, to study privacy-preserving big data analysis techniques in our future work with specific regard to improving S$k$LE/S$k$SE and proposing a privacy-preserving secure maximum/minimum protocol. By applying these protocols to big data analysis techniques like clustering, we will to contribute research on efficient and privacy-preserving big data analysis.

**Author Contributions:** Conceptualization, J.P.; Methodology, J.P.; Software, J.P.; Validation, D.H.L.; Formal analysis, J.P.; Writing—original draft, J.P.; Writing—review & editing, D.H.L.; Project administration, D.H.L.; Funding acquisition, D.H.L. All authors have read and agreed to the published version of the manuscript.

**Funding:** This research was partly supported by Basic Science Research Program through the National Research Foundation of Korea(NRF) funded by the Ministry of Education (NRF-2022R1A6A3A01087466) and Institute of Information & communications Technology Planning & Evaluation(IITP) grant funded by the Korea government(MSIT) (No.2021-0-00518, Blockchain privacy preserving techniques based on data encryption).

**Institutional Review Board Statement:** Not applicable.

**Informed Consent Statement:** Not applicable.

**Data Availability Statement:** Not applicable.

**Conflicts of Interest:** The authors declare no conflict of interest.

## Appendix A. Secure Zero Protocol (SZP)

In this section, we explain secure zero protocol (SZP) to privately compute whether all input data are zero or not. SZP is a real protocol to privately compute $F_{SZP}$ defined in Section 4.1, and it is constructed based on the idea of the existing equality protocol. We present SZP in Algorithm A1 and provide an example in Table A1 for easy understanding.

---

**Algorithm A1:** Secure Zero Protocol (SZP)

---

**DH input** : $\{E(x_i)\}_{i\in[l]}$

**CSP input** : secret key *SK*

**DH output**: $E(\gamma) = \begin{cases} E(1), \text{ if } x_i = 0 \text{ for all } i \in [l] \\ E(0), \text{ otherwise} \end{cases}$

**1 DH:**

**2**    toss a random coin $\alpha \in_R \{0,1\}$

**3**    **if** $\alpha = 0$ **then**

**4**      **for** $i \leftarrow 1,\ldots,l$ **do**

**5**        $E(s_i) \leftarrow E(-1) * E(x_i) * (\prod_{j=i+1}^{l} E(x_j))^2$

**6**        $E(c_i) \leftarrow E(s_i)^{r_i}$ where $r_i \neq 0 \in_R \mathbb{Z}_N$

**7**      **end**

**8**    **else**

**9**      $E(s_1) \leftarrow \prod_{i=1}^{l} E(x_i)$

**10**      $E(c_1) \leftarrow E(s_1)^{r_1}$ where $r_1 \neq 0 \in_R \mathbb{Z}_N$

**11**      **for** $i \leftarrow 2,\ldots,l$ **do**

**12**        $E(c_i) \leftarrow E(r_i)$ where $r_i \neq 0 \in_R \mathbb{Z}_N$

**13**      **end**

**14**    **end**

**15**    $\{d'_i\}_{i\in[l]} \leftarrow \sigma(\{E(c_i)\}_{i\in[l]})$

**16**    $DH \rightarrow CSP : \{d'_i\}_{i\in[l]}$

**17 CSP:**

**18**    $\{d_i\}_{i\in[l]} \leftarrow \{D(d'_i)\}_{i\in[l]}$

**19**    **if** $\exists d_i = 0 \text{ in } \{d_i\}_{i\in[l]}$ **then**

**20**      $E(\beta) \leftarrow E(0)$

**21**    **else**

**22**      $E(\beta) \leftarrow E(1)$

**23**    **end**

**24**    $CSP \rightarrow DH : E(\beta)$

**25 DH:**

**26**    **if** $\alpha = 0$ **then**

**27**      $E(\gamma) \leftarrow E(\beta)$

**28**    **else**

**29**      $E(\gamma) \leftarrow E(1) * E(\beta)^{N-1}$

**30**    **end**

**31 return** $E(\gamma)$

---

**Table A1.** Example of Algorithm A1 for SZP ($l = 6$).

| Input | $i$ | **1** | **2** | **3** | **4** | **5** | **6** | $i$ | **1** | **2** | **3** | **4** | **5** | **6** |
|---|---|---|---|---|---|---|---|---|---|---|---|---|---|---|
| | | | $F : \forall x_i = 0\ (\alpha = 0)$ | | | | | | | $F : \exists x_i \neq 0\ (\alpha = 1)$ | | | | |
| (Case 1) $\forall x_i = 0$ | $E(x_i)$ | $E(0)$ | $E(0)$ | $E(0)$ | $E(0)$ | $E(0)$ | $E(0)$ | $E(x_i)$ | $E(0)$ | $E(0)$ | $E(0)$ | $E(0)$ | $E(0)$ | $E(0)$ |
| | $E(s_i)$ | $E(-1)$ | $E(-1)$ | $E(-1)$ | $E(-1)$ | $E(-1)$ | $E(-1)$ | $E(s_1), E(c_1)$ | | | $E(0), E(0)$ | | | |
| | $E(c_i)$ | $E(r)$ | $E(r)$ | $E(r)$ | $E(r)$ | $E(r)$ | $E(r)$ | $E(c_i)$ | $E(0)$ | $E(r)$ | $E(r)$ | $E(r)$ | $E(r)$ | $E(r)$ |
| | $E(\beta)$ | | | $E(1)$ | | | | $E(\beta)$ | | | $E(0)$ | | | |
| | $E(\gamma)$ | | | $E(1)$ | | | | $E(\gamma)$ | | | $E(1)$ | | | |
| (Case 2) $\exists x_i = 1$ | $E(x_i)$ | $E(0)$ | $E(0)$ | $E(1)$ | $E(0)$ | $E(1)$ | $E(0)$ | $E(x_i)$ | $E(0)$ | $E(0)$ | $E(1)$ | $E(0)$ | $E(1)$ | $E(0)$ |
| | $E(s_i)$ | $E(3)$ | $E(3)$ | $E(2)$ | $E(1)$ | $E(0)$ | $E(-1)$ | $E(s_1), E(c_1)$ | | | $E(2), E(r)$ | | | |
| | $E(c_i)$ | $E(r)$ | $E(r)$ | $E(r)$ | $E(r)$ | $E(0)$ | $E(r)$ | $E(c_i)$ | $E(r)$ | $E(r)$ | $E(r)$ | $E(r)$ | $E(r)$ | $E(r)$ |
| | $E(\beta)$ | | | $E(0)$ | | | | $E(\beta)$ | | | $E(1)$ | | | |
| | $E(\gamma)$ | | | $E(0)$ | | | | $E(\gamma)$ | | | $E(0)$ | | | |
| (Case 3) $\forall x_i = 1$ | $E(x_i)$ | $E(1)$ | $E(1)$ | $E(1)$ | $E(1)$ | $E(1)$ | $E(1)$ | $E(x_i)$ | $E(1)$ | $E(1)$ | $E(1)$ | $E(1)$ | $E(1)$ | $E(1)$ |
| | $E(s_i)$ | $E(10)$ | $E(8)$ | $E(6)$ | $E(4)$ | $E(2)$ | $E(0)$ | $E(s_1), E(c_1)$ | | | $E(6), E(r)$ | | | |
| | $E(c_i)$ | $E(r)$ | $E(r)$ | $E(r)$ | $E(r)$ | $E(r)$ | $E(0)$ | $E(c_i)$ | $E(r)$ | $E(r)$ | $E(r)$ | $E(r)$ | $E(r)$ | $E(r)$ |
| | $E(\beta)$ | | | $E(0)$ | | | | $E(\beta)$ | | | $E(1)$ | | | |
| | $E(\gamma)$ | | | $E(0)$ | | | | $E(\gamma)$ | | | $E(0)$ | | | |

$r$: random value.

Intuitively, DH selects functionality $F$ ($F : \forall x_i = 0$ or $F : \exists x_i \neq 0$) by a random coin $\alpha$ where $F : \forall x_i = 0$ computes whether or not all input data $\{x_i\}_{i \in [l]}$ are 0, and $F : \exists x_i \neq 0$ computes whether or not there are any non-zero $x_i$ at least once in an input dataset $\{x_i\}_{i \in [l]}$. DH computes the selected functionality $F$ on an input dataset and sends CSP the computation result, which is a vector with either random values including 0 or only random values. CSP converts the vector and returns the converted value ($\beta$) back to DH. Then, DH outputs the result ($\gamma$) based on the converted value according to the random coin $\alpha$ (functionality $F$) selected by DH.

For easy understanding of Algorithm A1, we intuitively explain the data without encryption. DH selects functionality $F$ by tossing a random coin $\alpha$ (line 2) where $F : \forall x_i = 0$ if $\alpha = 0$; otherwise, $F : \exists x_i \neq 0$. When DH selects $\alpha = 0$ (resp., $\alpha = 1$), if all input data $\{x_i\}_{i \in [l]}$ are 0, DH sends CSP a vector $c$ that consists of only random values (resp., random values including 0) so that CSP returns $\beta = 1$ (resp., $\beta = 0$) back to DH. If $x_i = 1$ exists at least once in an input dataset $\{x_i\}_{i \in [l]}$, DH sends CSP a vector $c$ with random values including 0 (resp., only random values) so that CSP returns $\beta = 0$ (resp., $\beta = 1$) back to DH. Specifically, when DH selects a random coin $\alpha = 0$ ($F : \forall x_i = 0$), it computes $s_i$ on an input data $x_i$ for $i = 1, \ldots, l$ as follows.

$$s_i \leftarrow -1 + x_i + 2 \sum_{j=i+1}^{l} x_j \tag{A1}$$

If all input data $\{x_i\}_{i \in [l]}$ are 0, $c$ is a vector with only random values since all $s_i = -1$. When $x_i = 1$ exists at least once in an input dataset $\{x_i\}_{i \in [l]}$, let the last position with $x_i = 1$ be the $t$-th bit. $c_t = 0$ since $s_t = 0$ and the other $c_i$ for $i \neq t$ are all random since the other $s_i \neq 0$ for $i \neq t$, and therefore, $c$ is a vector that consists of random values including 0 (lines 4–7). When DH selects a random coin $\alpha = 1$ ($F : \exists x_i \neq 0$), $s_1$ is the sum of all input data $\{x_i\}_{i \in [l]}$ (line 9). If all input data $\{x_i\}_{i \in [l]}$ are 0, both $s_1$ and $c_1$ are 0 (lines 9–10), and therefore $c$ is a vector that consists of random values including 0 (lines 11–13). If $x_i = 1$ exists at least once in an input dataset $\{x_i\}_{i \in [l]}$, $c$ is a vector with only random values since $c_1$ is random (lines 9–13). After permutation $\sigma$ of $\{c_i\}_{i \in [l]}$, DH sends CSP the permutated

vector $d$ (lines 15–16). The subsequent process is the same as Algorithm 1 of SCI protocol. CSP then sends DH $\beta = 0$ (resp., $\beta = 1$) if a vector received from DH contains random values including 0 (resp., only random values) (lines 18–24). Then, the result $\gamma$ is $\beta$ if $\alpha = 0$; otherwise, the complement of $\beta$ (lines 26–30).

Similar to the Algorithm 1 of the SCI protocol, the result $\gamma$ of SZP satisfies the condition $\gamma = (\forall x_i = 0) = (\alpha \oplus \beta)$ for an input dataset $\{x_i\}_{i \in [l]}$. Since DH cannot know $\beta$ and CSP cannot learn information about $\alpha$, SZP does not disclose any information about the result. Specifically, DH knows a random coin $\alpha$ but cannot learn information about $\beta$ since the value $\beta$ is encrypted with a semantically secure encryption scheme. Likewise, CSP learns $\beta$ by decryption but cannot learn information about $\alpha$ since the intermediate result from DH is blinded by a random value and the value $\alpha$ is chosen uniformly at random. In this paper, we do not include a proof for SZP security.

*Computation and communication costs*: SZP requires $l$ encryptions/decryptions and at most $(l + 1)$ exponential computations, and $(l + 1) \cdot C$ bits are transmitted in one round. Specifically, CSP decrypts $d'_i$ for $i = 1, \ldots, l$ in line 18. Encryption for $E(\beta)$ in line 20 and 22 is excluded since it can be executed by precomputation. According to $\alpha$, exponential computations are executed $l$ or one times in lines 3–14, and at most one time in lines 26–30.

### Appendix B. Privacy-Preserving $k$-Nearest Neighbor Classification (P$k$NC)

In this section, we present P$k$NC in Algorithm A2, which privately classifies an unclassified input query based on a classified dataset. In order to construct P$k$NC, additional functionalities are required. For information on the detailed protocols, refer to [10].

---

**Algorithm A2:** Privacy-preserving $k$-Nearest Neighbor Classification (P$k$NC)

**DH input** : dataset $\{E(d_{i,j}), E(c_i)\}_{i \in [n], j \in [m]}$
　　　　　　　input query $\{E(q_j)\}_{j \in [m]}$
**CSP input** : secret key $SK$
**DH output**: input query $q$ is classified into the $\alpha$-th class if $K'_\alpha = 1$ in $\{E(K'_j)\}_{j \in [t]}$.

1 　DH and CSP run interactive protocols with third parties to ideally compute
　　functionalities.
2 　**for** $i \leftarrow 1, \ldots, n$ **do**
3 　　$\quad$ $(E(e_i), \perp) \leftarrow F_{SSED}(\{E(d_{i,j}), E(q_j)\}_{j \in [m]}, SK)$
4 　　$\quad$ $(\langle E(\overline{e_i}) \rangle_B, \perp) \leftarrow F_{SBD''}(E(e_i), SK)$
5 　**end**
6 　$(\{E(K_i)\}_{i \in [n]}, \perp) \leftarrow F_{SkLE_S}(\{\langle E(\overline{e_i}) \rangle_B\}_{i \in [n]}, SK)$
7 　$(\{E(f_j)\}_{j \in [t]}, \perp) \leftarrow F_{SCF}(\{E(c_i), E(K_i)\}_{i \in [n]}, SK)$
8 　**for** $j \leftarrow 1, \ldots, t$ **do**
9 　　$\quad$ $(\langle E(f_j) \rangle_B, \perp) \leftarrow F_{SBD'}(E(f_j), SK)$
10 　**end**
11 　$(\{E(K'_j)\}_{j \in [t]}, \perp) \leftarrow F_{S1LE_S}(\{\langle E(f_j) \rangle_B\}_{j \in [t]}, SK)$
12 　**return** $\{E(K'_j)\}_{j \in [t]}$

---

*Secure Squared Euclidean Distance functionality* $F_{SSED}$: $F_{SSED}$ privately computes the squared Euclidean distance for two input data with $m$ attributes. $F_{SSED}$ receives $\{E(a_j), E(b_j)\}_{j \in [m]}$ from DH and a secret key $SK$ from CSP, and then it sends $E(e)$ to DH where $e = \sum_{j=1}^{m}(a_j - b_j)^2$.

*Secure Class Frequency functionality* $F_{SCF}$: Given the class information $(c_i)$ of $k$ data most similar to an input query, $F_{SCF}$ privately computes the number for each class of the $k$ data. $F_{SCF}$ receives $\{E(c_i), E(K_i)\}_{i \in [n]}$ from DH and a secret key $SK$ from CSP, and then it sends $\{E(f_j)\}_{j \in [t]}$ to DH where $f_j$ is the number of $j$-th classes in $k$ data most similar to an input query.

*Other functionalities $F_{SBD'}$, $F_{SBD''}$, and $F_{S1LE}$*: We introduced $F_{SBD}$ in Section 3.6 where, given an encrypted data $E(x)$, $F_{SBD}$ returns ciphertexts for the individual bits and their 1's complement of corresponding data $x$ (i.e., $\langle E(x)\rangle_B, \langle E(\bar{x})\rangle_B$). Similarly, $F_{SBD'}$ (with single quotation mark) returns ciphertexts $\langle E(x)\rangle_B$ for the individual bits of corresponding data $x$, and $F_{SBD''}$ (with double quotation mark) returns ciphertexts $\langle E(\bar{x})\rangle_B$ for 1's complements of corresponding data $x$. $F_{S1LE}$ means $F_{SkLE}$ with $k = 1$, which privately computes the maximum data in a dataset. In other words, data $e_i$ with $K_i = 1$ is the maximum.

We assume that a classified dataset consists of $n$ data, and their classes $\{d_i, c_i\}_{i\in[n]}$ where data $d_i$ consists of $m$ attributes (i.e., $d_i = \{d_{i,j}\}_{j\in[m]}$). Similar to data, we assume that an input query $q$ consists of $m$ attributes (i.e., $q = \{q_j\}_{j\in[m]}$). We assume that DH has an encrypted dataset $\{E(d_{i,j}), E(c_i)\}_{i\in[n],j\in[m]}$ and an encrypted input query $\{E(q_j)\}_{j\in[m]}$. After P$k$NC, DH returns the class information of the input query based on the dataset in an encrypted form. Specifically, DH returns $\{E(K'_j)\}_{j\in[t]}$ after P$k$NC, and if $K'_\alpha = 1$, the input query is classified into the $\alpha$-th class. Recall that $n$ is the number of data, $m$ is the number of attributes, and $t$ is the number of class types. For easy understanding of Algorithm A2, we intuitively explain the data without encryption.

*(Step 1: line 3) privately computing distances between an input query and data in a dataset*: $F_{SSED}$ privately computes the squared Euclidean distance $e_i$ between an unclassified input query $q$ and the data $d_i$ in a dataset.

*(Step 2: lines 4–6) privately selecting $k$ smallest distances*: In line 6, $F_{SkLE_S}$, whose input is 1's complement of distances, privately computes $k$ data closest to an input query as mentioned in Section 4.3. In other words, it privately computes $k$ smallest distances between an input query and data in a dataset. $F_{SBD''}$ is used to comply with the input format of $F_{SkLE_S}$.

*(Step 3: lines 7–11): privately computing the majority class of $k$ data corresponding to the $k$ smallest distances*: $F_{SCF}$ privately computes the number of each class of $k$ data closest to an input query. $F_{S1LE}$ privately computes the majority class of the $k$ data by computing the maximum number of the classes where $F_{SBD'}$ is used to comply with the input format of $F_{S1LE}$.

P$k$NC discloses no information since DH receives only semantically secure ciphertexts from third parties to ideally compute functionalities. Similar to the formal proof for S$k$LE$_S$ in Section 4.4, we can simulate DH's view in random values since DH receives only randomized ciphertexts from third parties for functionalities. Since CSP does not receive any messages, we do not consider CSP's view. We do not include a formal proof of P$k$NC security in this paper.

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
