# Peer review of "Parallelly Running and Privacy-Preserving k-Nearest Neighbor Classification in Outsourced Cloud Computing Environments"

_electronics, doi:10.3390/electronics11244132_

Round 1

Reviewer 1 Report

The authors propose a privacy-preserving k-nearest neighbor classification in outsourced cloud computing environments, the following observations were made in the manuscript :

Comments:

1.       The novelty and contribution of the paper should be made clearer. Specifically, the method presented in this paper is not an improvement over [10] since the trade-off between processing speed and security in terms of data size remains the same exchange. We can still interpret the data size disclosure as not posing a significant security risk to the research [10]. If this is not well explained it will make this paper look like a minor revision of [10] and will reduce the contribution of the paper quite a bit.

2.       The authors should compare with state-of-the-art approachs, research [14] has been presented since 2014.

3.       The article should be reorganized to make it easier for readers to follow, for as by placing the "Related Section" section after the "Introduction" section.

Author Response

We appreciate the reviewers’ comments and an opportunity of resubmission. With many helpful comments, we improved the prior study and revised our manuscript. We uploaded the response of your comments as a file.

Reviewer 2 Report

In this study, the authors propose a new secure comparison and SkLE/SkSE protocols to solve the abovementioned information disclosure problems and implement PkNC with these novel protocols. The idea seams interesting and novel. However, I have following suggestions to improve its overall quality.

 1.      The section” Contributions” should be simplified to express several contributions of this paper directly, and the Fig 1. should not be placed in this section, but in the experimental section.

2.      It is recommended to use a diagram to clearly represent the “System Model”.

3.      “At a high level, DH selects functionality…..”. What is a high level? A specific explanation is needed.

4.      “We assume that πSkLES computes auxiliary data for the j-th bit of all elements….”. The reasonableness of this assumption requires further explanation.

5.      In the same field of interest there are some latest papers that would increase the technical strength of the article like, - LocJury: An IBN-Based Location Privacy Preserving Scheme for IoCV. IEEE Trans. Intell. Transp. Syst. 22(8): 5028-5037 (2021), - An IoT data sharing privacy preserving scheme. INFOCOM Workshops 2020: 984-990, - PROCESS: Privacy-Preserving On-Chain Certificate Status Service. INFOCOM 2021: 1-10, - Efficient and secure k-nearest neighbor query on outsourced data[J]. Peer-to-Peer Networking and Applications, 2020(11). - An Efficient Filter-Based Feature Selection Model to Identify Significant Features from High-Dimensional Microarray Data. Arabian Journal for Science and Engineering. Section A, Sciences (2020).

Author Response

(The authors gave the same response as above.)

Round 2

Reviewer 1 Report

The authors had addressed all my concerns.

Author Response

Thank you for your review